# THE TRICKLE-DOWN IMPACT
# OF REWARD (IN-)CONSISTENCY ON RLHF

**Lingfeng Shen**♣∗ **Sihao Chen**♠ **Linfeng Song**♡
**Lifeng Jin**♡ **Baolin Peng**♡ **Haitao Mi**♡ **Daniel Khashabi**♣ **Dong Yu**♡
♣Johns Hopkins University ♠University of Pennsylvania ♡Tencent AI Lab

## ABSTRACT

Standard practice within Reinforcement Learning from Human Feedback (RLHF) involves optimizing against a Reward Model (RM), which itself is trained to reflect human preferences for desirable generations. A notable subject that is understudied is the *(in-)consistency* of RMs — whether they can recognize the semantic changes to different prompts and appropriately adapt their reward assignments — and their impact on the downstream RLHF model.

In this paper, we visit a series of research questions relevant to RM inconsistency: (1) How can we measure the consistency of reward models? (2) How consistent are the existing RMs and how can we improve them? (3) In what ways does reward inconsistency influence the chatbots resulting from the RLHF model training?

We propose CONTRAST INSTRUCTIONS – a benchmarking strategy for the consistency of RM. Each example in CONTRAST INSTRUCTIONS features a pair of lexically similar instructions with different ground truth responses. A consistent RM is expected to rank the corresponding instruction and response higher than other combinations. We observe that current RMs trained with the standard ranking objective fail miserably on CONTRAST INSTRUCTIONS compared to average humans. To show that RM consistency can be improved efficiently without using extra training budget, we propose two techniques CONVEXDA and REWARDFUSION, which enhance reward consistency through extrapolation during the RM training and inference stage, respectively. We show that RLHF models trained with a more consistent RM yield more useful responses, suggesting that reward inconsistency exhibits a trickle-down effect on the downstream RLHF process.

## 1 INTRODUCTION

Recently, *reinforcement learning from human feedback* (RLHF) has emerged as a popular technique to optimize and align a language model with human preferences (Ouyang et al., 2022). RLHF provides a natural solution for optimizing non-differentiable, scalar objectives for language models, and has been the centerpiece of recent state-of-the-art large language models (LLMs) (Lu et al., 2022; Hejna III & Sadigh, 2023; Go et al., 2023; Korbak et al., 2023; OpenAI, 2023).

In RLHF, a *reward model* (RM) generates scalar rewards for model-generated outputs as supervision during reinforcement learning. RMs are typically calibrated to proxy human preferences/rankings of responses, in the context of an input instruction. Since policy gradient methods optimize based on this reward function, the reward function inevitably dictates the behavior of the resultant chatbot. As such, the properties of RMs and their impact on RLHF models have become points of interest for the research community (Gao et al., 2022; Zhu et al., 2023; Dong et al., 2023).

In this work, we study the phenomenon of *reward inconsistency* in RMs, i.e., current RMs trained with the standard ranking objective on human preference data (§2) often fail to distinguish between more vs. less favorable responses with respect to real-world instructions. We observe that reward inconsistency has a *trickle-down effect* on the RLHF process — the more inconsistent the RM is, the more likely the resulting chatbot is to generate inaccurate or less useful responses.

---

∗Most of the work done while Lingfeng and Sihao were interns at the Tencent AI Lab.

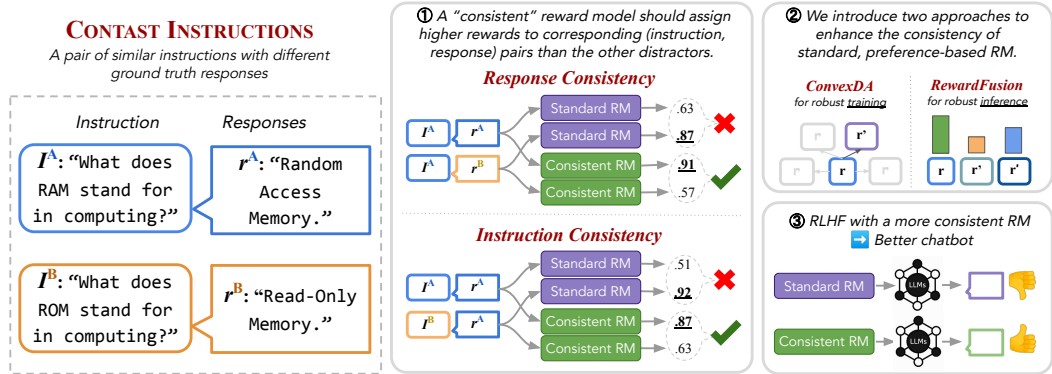

Figure 1: We introduce CONTRAST INSTRUCTIONS as a way to measure the *consistency* of a reward model. Each example consists of two similar instructions with different ground truth responses. Given one of the instructions or responses, whether or not an RM could rank its corresponding response or instruction higher than the other distractor indicates the consistency of the RM.

To illustrate and quantify the degree of reward inconsistency in RMs, we propose CONTRAST IN-STRUCTIONS, a simple, intuitive benchmarking strategy that can be used with any instruction-tuning or human preference dataset in a fully automated manner. CONTRAST INSTRUCTIONS involves pairs of similar instructions with different responses. As an example, consider two similar-looking prompts ($I^A$ and $I^B$) one about "RAM" and "ROM" (Figure 1; left). Despite the similarity of these prompts, they warrant different responses. Our consistency metrics measure whether a reward model can appropriately identify the correspondence between prompts and responses. Specifically, given a pair of such (instruction, response) examples, we consider an RM *consistent* if it assigns a higher score to the corresponding (instruction, response) compared to the other combinations, i.e. swapping instructions or responses between the two examples, as we show with ① in Figure 1.

We construct CONTRAST INSTRUCTIONS with four popular open-source human preference or instruction-tuning datasets. Surprisingly, we observe close to random-chance performance when we evaluate standard RMs trained with ranking objectives (e.g. LLaMa-7B) on CONTRAST IN-STRUCTIONS, while humans are able to rank the responses correctly in $\approx 80\%$ of the cases. The performance gap indicates the inherent reward inconsistency from standard RM training and inference.

To enhance the consistency of standard RM, we introduce two techniques (§5) : **CONVEXDA** and **REWARDFUSION** (② in Figure 1). The two methods can be incorporated during RM training and inference, respectively, without incurring extra computational costs during training. While our experimental results indicate some improvements, the gap with respect to human performance remains large. Interestingly, our analysis (§6) reveals that using a more consistent RM during RLHF training leads to more useful responses from the downstream RLHF model (③ in Figure 1). Our findings suggest the value of reward (in-)consistency as an intrinsic evaluation metric for preference-based RMs, potentially enabling easier access for future research on reward modeling and RLHF.

The main contributions of this paper are:

- We introduce CONTRAST INSTRUCTIONS, an intuitive yet scalable benchmarking strategy for evaluating RM consistency. We observe a wide performance gap between standard, preference-based RM vs. human judgments, which suggests sizable room for improvements in standard reward modeling and evaluation strategy.

- We show that reward consistency can be enhanced without extra training costs. We demonstrate this with two techniques **CONVEXDA** and **REWARDFUSION**, which can be applied during RM training and inference stages, respectively.

- We empirically show that training RLHF models with more consistent RMs would result in the RLHF model generating more useful responses. We provide thorough analysis and examples, which help us understand the advantage of a more consistent RM over a less inconsistent one.

## 2 PRELIMINARIES: REWARD MODELING FOR RLHF

**Reward model.** Following the conventional setup (Ziegler et al., 2019), we define RM $\mathcal{R}_\theta$ as a scalar function that allows us to train a generative model during RLHF. To supervise RM, we are given a dataset of human preferences $\mathcal{D}_h$. Each instance in this dataset $(I_i, r_i^+, r_i^-)$ is comprised of an instruction prompt $I_i$, a pair of responses $r_i^+, r_i^-$ where $r_i^+$ is preferred over $r_i^-$ by humans. On this labeled data, $\mathcal{R}_\theta$ is trained to assign a higher scalar reward to human-preferred $r_i^+$ over non-preferred $r_i^-$ in the context of $I_i$ This can be achieved by minimizing the ranking loss $\mathcal{L}$, where $\sigma$ is the sigmoid function and $I_i \circ r_i^+$ is the concatenation of $I_i$ and $r_i^+$.

$$\mathcal{L}(\theta) = -\mathbb{E}_{(I_i, r_i^+, r_i^-) \sim \mathcal{D}_h} \left[ \log \left( \sigma \left( \mathcal{R}_\theta(I_i \circ r_i^+) - \mathcal{R}_\theta \left( I_i \circ r_i^- \right) \right) \right) \right]. \tag{1}$$

**Reinforcement Learning.** The last stage of RLHF is reinforcement learning. Specifically, a per-token KL penalty from the supervised fine-tuning (SFT) model at each token to mitigate over-optimization of the reward model, and the value function is initialized from the RM. We maximize the following combined objective function $\mathcal{J}(\phi)$ in RL training based on PPO algorithm (Schulman et al., 2017; Ouyang et al., 2022), RL training dataset $\mathcal{D}_{\pi_\phi^{\text{RL}}}$ and pre-training dataset $\mathcal{D}_{\text{pre}}$:

$$\mathcal{J}(\phi) = \mathbb{E}_{(I,r) \sim \mathcal{D}_{\pi_\phi^{\text{RL}}}} \left[ \mathcal{R}_\theta(I \circ r) - \beta \log \left( \pi_\phi^{\text{RL}}(r \mid I) / \pi^{\text{SFT}}(r \mid I) \right) \right] + \gamma \mathbb{E}_{r \sim \mathcal{D}_{\text{pre}}} \left[ \log \left( \pi_\phi^{\text{RL}}(r) \right) \right],$$

where $\mathcal{R}_\theta(I \circ r)$ is the reward function, and $\pi_\phi^{\text{RL}}$ is the learned RL policy parameterized by $\phi$ initialized from a pretrained supervised trained model $\pi^{\text{SFT}}$. The first term encourage the policy $\pi_\phi^{\text{RL}}$ to generate responses that have higher reward scores. The second term represents a per-token KL reward controlled by coefficient $\beta$ between $\pi_\phi^{\text{RL}}$ and $\pi^{\text{SFT}}$ to mitigate over-optimization toward the reward model during RL training. The third optional term provides regularization by encouraging responses with high probability from the pretraining dataset with coefficient $\gamma$.

## 3 CONTRAST INSTRUCTIONS: MEASURING REWARD (IN-)CONSISTENCY

Conventionally in RLHF, reward models are trained explicitly to distinguish the more vs. less favorable responses in the context of an instruction (Eq. 1). One would expect an RM to consistently predict higher reward scores toward the more favorable instruction-response pairings. For instance, as the example in Figure 1 shows, an ideal reward should assign a higher score to $r_A$ appearing in response to $I_A$, than $I_B$. In practice, however, RMs usually suffer from over-optimization towards the training distribution, leading to *inconsistencies* between RM predictions vs. human preferences at inference time (Gao et al., 2022).

To illustrate and measure reward inconsistency in RMs, we introduce CONTRAST INSTRUCTIONS. A CONTRAST INSTRUCTIONS benchmark takes the form of $\mathcal{D} = \{(I_i^A, I_i^B, r_i^A, r_i^B)\}_{i=1}^N$ (see Fig. 1). Each test instance consists of instruction-response pairs $(I^A, r^A)$ and $(I^B, r^B)$. To make the benchmark meaningfully challenging, we sample $I^A$ and $I^B$ such that the two are lexically similar instructions with different semantics (details later in §3.1). $r^A$ and $r^B$ are the human-preferred responses to $I^A$ and $I^B$, respectively. Conceptually, a consistent RM should be able to identify pairs of corresponding instruction-response. Concretely, this means assuming higher score to $(I^A, r^A)$ than $(I^A, r^B)$ or $(I^B, r^A)$. There are two ways one can quantify such notions of reward consistency:

- **Response Consistency ($\mathcal{C}_{\text{res}}$):** Given one of the instructions $I^A$, we measure if a RM ($\mathcal{R}_\theta$) can identify the corresponding response $r^A$ by assigning higher rewards to $r^A$ over $r^B$.

$$\mathcal{C}_{\text{res}} = \frac{1}{|\mathcal{D}|} \sum_{(I_i^A, I_i^B, r_i^A, r_i^B) \in \mathcal{D}} \mathbf{1} \left[ \mathcal{R}_\theta \left( I_i^A \circ r_i^A \right) > \mathcal{R}_\theta \left( I_i^A \circ r_i^B \right) \right]. \tag{2}$$

- **Instruction Consistency ($\mathcal{C}_{\text{ins}}$):** Similarly, given one of the responses $r^A$, we measure if a RM ($\mathcal{R}_\theta$) can assign higher rewards to its corresponding instruction $I^A$ over the distractor $I^B$.

$$\mathcal{C}_{\text{ins}} = \frac{1}{|\mathcal{D}|} \sum_{(I_i^A, I_i^B, r_i^A, r_i^B) \in \mathcal{D}} \mathbf{1}\left[\mathcal{R}_\theta\left(I_i^A \circ r_i^A\right) > \mathcal{R}_\theta\left(I_i^B \circ r_i^A\right)\right]. \tag{3}$$

It is worth noting that with the standard learning objective (Eq. 1), RMs are explicitly trained to rank responses but not instructions, and so $\mathcal{C}_{\text{res}}$ conceptually resembles the RM learning objective, while $\mathcal{C}_{\text{ins}}$ does not. Therefore, we expect the standard RM to perform better under the $\mathcal{C}_{\text{res}}$ metric compared to $\mathcal{C}_{\text{ins}}$, which we will further discuss in §4.

### 3.1 Constructing Contrast Instructions Automatically

CONTRAST INSTRUCTIONS can be automatically constructed from datasets that contain human preferences. Inspired by Gardner et al. (2020), we formulate examples in CONTRAST INSTRUCTIONS such that the pair of instructions $I^A$, $I^B$ are lexically similar, yet their ground truth responses are different. We expect a *consistent* RM to recognize the nuanced semantic difference between the instructions, and recognize the corresponding answer to an instruction.

**Benchmark Construction.** We adopt four open-source human preference datasets of various NLP tasks: STACKEXCHANGE for question answering (Askell et al., 2021), WMT for machine translation (Ma et al., 2019), REALSUMM for text summarization (Bhandari et al., 2020), and TWITTER for paraphrase generation (Shen et al., 2022c). Each dataset features examples of an instruction comprised of both a task prompt and a task-specific input, and the corresponding responses ranked by human preference. Within each dataset, we sample pairs of similar instructions with sentence embedding model SimCSE (Gao et al., 2021). To ensure the instruction pairs are similar but not semantically equivalent, we keep only instruction pairs with cosine similarity within $[0.75, 0.9]$. We show the statistics and examples of each resulting CONTRAST INSTRUCTIONS dataset in Table 1

| Data | Instructions (task prompt omitted) | Responses |
|---|---|---|
| STACK | $I^A$: What are the three primary colors in the subtractive color models? | $r^A$: The subtractive primaries are cyan, magenta, and yellow. |
| | $I^B$: What're the three primary colors in the additive color models? | $r^B$: They're red, green, and blue. |
| WMT | $I^A$: Mindestens 410 Menschen wurden bei einem durch ein starkes Erdbeben ausgelösten Tsunami in Indonesien getötet. | $r^A$: At least 410 people were killed in a tsunami triggered by a strong earthquake in Indonesia. |
| | $I^B$: Das Erdbeben in der Provinz Zentralsulawesi hat 410 Menschen getötet. | $r^B$: The earthquake in Central Sulawesi province has killed 410 people. |
| TWITTER | $I^A$: But my bro from the 757 EJ Manuel is the 1st QB gone. | $r^A$: My boy EJ Manuel being the 1st QB picked. |
| | $I^B$: EJ Manuel will be the 1st QB taken in the 2013 NFL Draft. | $r^B$: EJ Manuel selected as the 1st QB in the 2013 NFL Draft. |
| REALSUMM | $I^A$: Mary Ann Diano was left homeless and hopeless when the storms hit Staten Island, New York, in October 2012. (...) | $r^A$: Mary Ann Diano, 62, lost her home in Staten Island in October 2012.. (...) |
| | $I^B$: Denise and Glen Higgs, from Braunton, Devon, had all but lost hope that they would ever be able to conceive after glen was made infertile due to cancer treatment. (...) | $r^B$: Denise and Glen Higgs lost hope after glen was made infertile due to cancer treatment, but the couple had a daughter Mazy after IVF treatment. (...) |

Table 1: Examples of the four CONTRAST INSTRUCTIONS benchmark datasets.

## 4 Inconsistency of existing reward models

With CONTRAST INSTRUCTIONS, we evaluate the consistency of RMs trained with the standard ranking objective (Eq. 1).

**Experimental setup.** We initialize RMs from `LLaMa-7B` checkpoint (Touvron et al., 2023) and finetune on each of the four human preference datasets used to create our CONTRAST INSTRUCTIONS benchmark. We also finetune a multi-task version (Mishra et al., 2022) on the mixture of four human preference datasets. We use the following configurations for RM training for both single-task and multi-task settings. Due to resource constraints, we adopt the Low-Rank Adaptor (LoRA) (Hu et al., 2021) for training. We use the AdamW optimizer and set a learning rate of 2e-5. For multi-task training, we combine the training set from selected benchmarks and train using LoRA with a learning rate of 3e-5. Finally, we report human performance resulting from the majority vote of three human annotators (the authors) on 100 randomly selected data points.

**Inconsistency of RMs on CONTRAST INSTRUCTIONS.** Table 2 summarizes the results of the fine-tuned RMs averaged on the four CONTRAST INSTRUCTIONS benchmarks. We observe that with a relatively large 7B parameter RM, it still performs close to random guessing in terms of both responses ($\mathcal{C}_{\text{res}}$) and instruction consistency ($\mathcal{C}_{\text{ins}}$). This trend can be seen on per-dataset as well, in Table 3. In our setting, multi-task training does not seem to help (no cross-task transfer), possibly due to the limited commonalities among these tasks. The reward models are slightly better in terms of $\mathcal{C}_{\text{res}}$ compared to $\mathcal{C}_{\text{ins}}$, which fits our expectation – the RM learning objective (Eq.1) is more similar to $\mathcal{C}_{\text{res}}$ than it is to $\mathcal{C}_{\text{ins}}$.

| Model | $\mathcal{C}_{\text{res}}$ | $\mathcal{C}_{\text{ins}}$ |
|---|---|---|
| Random | 50.0 | 50.0 |
| `LLaMa-7B` (Single-Task) | 53.6 | 49.4 |
| `LLaMa-7B` (Multi-Task) | 53.0 | 48.8 |
| Human (Estimated) | 82.8 | 81.7 |

Table 2: $\mathcal{C}_{\text{res}}$ and $\mathcal{C}_{\text{ins}}$ (Eq.2, Eq.3) averaged across four benchmarks. Standard RM training performs near random performance, indicating a major inconsistency gap between fine-tuned reward models and humans.

The estimated human performance shows a wide gap compared to the RM performance on CONTRAST INSTRUCTIONS, suggesting reward inconsistency can be attributed to the standard practice of reward modeling.

## 5    ENHANCING REWARD MODEL CONSISTENCY

Having identified in §4 the inconsistencies of reward models, we now present methods to address these issues. Specifically, we propose two solutions: one to be applied during training (§5.1) and another during inference (§5.2), which does not impose additional computing costs. It's important to note that these techniques are designed to be agnostic to CONTRAST INSTRUCTIONS's format and setup, thereby minimizing the negative impact of Goodhart's law (Manheim & Garrabrant, 2018).

### 5.1    CONSISTENCY-INDUCING TRAINING VIA **CONVEXDA**

To mitigate the effect of over-optimization during RM training, we design **CONVEXDA**, a lightweight, efficient data augmentation technique, that neither modifies the RM learning objective nor increases the overall training cost. The high-level idea is to create various perturbations of the original input through data augmentation and select the most representative one to replace the original input example during training.

**Approach.** Given a human preference example $(I, r^+, r^-)$, we use an off-the-shelf textual data augmentation tool (Ma, 2019) to substitute words in the responses with synonyms according to WordNet (Miller, 1995) or PPDB (Ganitkevitch et al., 2013). For each example, we generate $N = 5$ augmented versions $\{I, r_j^+, r_j^-\}_{j=1}^N$. As using all $N$ data points for training would incur extra cost, we draw inspiration from previous studies (Chen et al., 2010; Sener & Savarese, 2018; Rajput et al., 2019; Agarwal et al., 2020) and select one data point among five that serve as a vertex of a convex hull in the embedding space. This geometric approach ensures that we focus on the most critical points within the set of augmented data points. The strategy keeps the training efficiency on par with the standard RM training while offering the advantages of data augmentation.

We use the SimCSE model (Gao et al., 2021) to embed each original and augmented response to a 768-dimensional vector. In principle, to construct a convex hull in a $k$-dimensional embedding space, we need at least $k+1$ data points. For such reason, we apply Principal Component Analysis (PCA) to reduce the dimensional of the embedding to 2, We identify and select the example that act

| METRIC | MODEL | METHOD | STACK | WMT | TWITTER | REALSUMM | AVG. |
|---|---|---|---|---|---|---|---|
| $\mathcal{C}_{\text{res}}$ | | Human | 81.0 | 89.0 | 83.0 | 78.0 | 82.8 |
| | LLaMa-7B | Standard RM | 50.1♠ | 56.4 | 60.1 | 47.7 | 53.6 |
| | | Our Approach | **53.6** | **60.4** | **62.9** | **52.6** | **57.3** |
| | | *Ablating "Our Approach"* | | | | | |
| | | – CONVEXDA | 52.1 (-1.5) | 58.4 (-2.0) | 61.7 (-1.2) | 50.2 (-2.4) | 55.6 (-1.7) |
| | | – REWARDFUSION | 53.1 (-0.5) | 59.9 (-0.5) | 62.1 (-0.8) | 51.2 (-1.4) | 56.5 (-0.8) |
| $\mathcal{C}_{\text{ins}}$ | | Human | 82.1 | 84.3 | 81.2 | 79.2 | 81.7 |
| | LLaMa-7B | Standard RM | 48.2♠ | 48.4 | 48.5 | 52.6 | 49.4 |
| | | Our Approach | **53.4** | **53.9** | 52.5 | **56.5** | **54.1** |
| | | *Ablating "Our Approach"* | | | | | |
| | | – CONVEXDA | 52.0 (-1.4) | 52.1 (-1.8) | **53.6** (+1.1) | 55.1 (-1.4) | 53.2 (-0.9) |
| | | – REWARDFUSION | 52.9 (-0.5) | 53.6 (-0.3) | 52.5 (0.0) | 55.7 (-0.8) | 53.7 (-0.4) |

Table 3: The evaluation of reward consistency from standard RM training vs. our approach across four different CONTRAST INSTRUCTIONS benchmarks, plus ablation studies on **CONVEXDA** and **REWARDFUSION**. ♠ indicates results evaluated on the official checkpoint from Stack-LLaMa;

as the corner points of a low-dimensional convex hull, and replace the original input with it during RM training.

## 5.2 CONSISTENCY-INDUCING INFERENCE VIA REWARDFUSION

During RM inference, we introduce **REWARDFUSION**. The method takes inspiration from (Zhao & Cho, 2019), where we first identify sentences similar to the response from the training corpus, and then use the weighted average reward score across the target response and the retrieved training samples as a better estimate for the reward score.

**Approach.** At the inference time, given a pair of instructions and response $I \circ r$, we again use the SimCSE model to retrieve a set of similar training examples $\{I \circ r^*\}$ that have cosine similarity to $I \circ r$ over a threshold $\delta$ from the training corpus. Then, we take the weighted average of the reward scores of the original plus retrieved training examples $X = (I \circ r) \cup \{I \circ r^*\}$:

$$\mathcal{R}_{\text{fusion}}(X) = \sum_{x \in X} \frac{\text{sim}(x, I \circ r)}{\sum_{x' \in X} \text{sim}(x', I \circ r)} \mathcal{R}_\theta(x). \tag{4}$$

The threshold $\delta = 0.95$ is selected based on averaged performance on development sets. The embedding retrieval process is instantiated with Faiss (Johnson et al., 2019).

## 5.3 EXPERIMENTS AND RESULTS

We conduct experiments to evaluate the effectiveness of both techniques in enhancing the consistency of RM on the four CONTRAST INSTRUCTIONS benchmarks. We follow the same experimental settings detailed in §4. We start with single-task RMs trained on each human preference dataset and apply **CONVEXDA** and **REWARDFUSION**.

**Findings.** We observe the following based on the results in Table 3. (1) Both **CONVEXDA** and **REWARDFUSION** effectively enhance the consistency of the reward model. (2) The combination of these two techniques works best, highlighting their complementarity in RM training and inference for enhancing consistency. (3) Despite the improvements brought by the two techniques, the overall performance on the CONTRAST INSTRUCTIONS benchmark remains limited. RMs with enhanced consistency from the two techniques still fall behind human performance by a large margin, demonstrating the challenges of addressing RM's consistency. We defer further details to Appendix F.

## 6 TRICKLE-DOWN EFFECT OF REWARD INCONSISTENTENCY ON RLHF

Next, we explore the merits of having a more consistent RM in downstream RLHF training by comparing RLHF-trained language models with a standard (§4) vs. more consistent RM (§5).

**Experimental setup.** We follow the overall experimental setup outlined in StackLLaMa (Beeching et al., 2023). For the experiments, we use `LLaMa-7B` to initialize the supervised finetuning (SFT), reward, and policy models. We train the models on StackExchange, which is segmented into SFT, RM, and RL datasets. More implementation details can be found Appendix C. We train two RLHF models, one with the standard RM, and the other with RM finetuned on examples with CONTRAST INSTRUCTIONS format sampled from StackExchange training split. We denote this as the CONTRAST Ft. RM. We conduct a human evaluation on 500 questions randomly sampled from the test split of StackExchange.

**Human evaluation.** To assess the general quality of the responses, we conduct two evaluations on responses generated by the two RLHF models. We first ask human raters to assess the *individual acceptability* of each model's response. This process involves a three-way judgment {`Accept`, `Reject`, `Unsure`}. A response is deemed acceptable only if it adequately addresses the

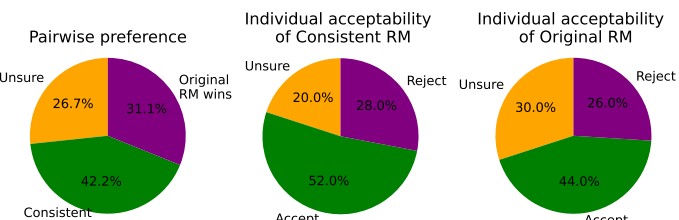

Figure 2: Human evaluation of the RLHF models trained with the consistent vs. original RM respectively. **Left** shows the pairwise preference, and **Middle / Right** shows individual acceptability of using consistent RM / original RM.

question in the prompt; exhibits no significant error; and contains no redundant information. In addition, we ask the human rater to annotate the *pairwise preference* between the two responses. We ask human raters to compare the outputs of two models and determine which model's output is preferred. We also provide raters the option to indicate a tie between the responses. The human evaluation results are shown in Figure 2. We observe that with the more consistent RM, the downstream RLHF model demonstrates higher generation quality.

**Automatic evaluation.** We use MT-bench[1] (Zheng et al., 2023) for automatic evaluation. MT-Bench is a benchmark featuring challenging ONE-TURN or MULTI-TURN reasoning and instruction-following examples, where the model responses are graded by GPT-4 on a scale of 1 (worst) to 10 (best). Overall, the results are shown in Table 4. We observe that more consistent RMs lead to RLHF models generating more preferable responses under both settings.

| Reward Model | One-turn | Multi-turn |
|---|---|---|
| Standard | 3.44 | 2.39 |
| CONTRAST Ft. | 3.89 | 3.18 |

Table 4: The evaluation results on the MT-BENCH (Zheng et al., 2023). Score ranges from 1 (worst) to 10 (best).

**Reward consistency ⇒ More useful responses.** To understand where the improvement lies, we follow criteria from Malaviya et al. (2023) and ask human raters to assess the *relevance, usefulness* and *factuality* of RLHF model responses. *Relevance* indicates whether the response is topically relevant to the instruction. *Usefulness* indicates whether a response serves as a useful and direct answer to the instruction. *Factuality* indicates the factu-

| Reward Model | Relevance | Usefulness | Factuality |
|---|---|---|---|
| Standard | 92.4% | 50.9% | 49.1% |
| CONTRAST Ft. | 92.2% | 57.8% | 50.0% |

Table 5: Fine-grained human evaluation of RLHF model response along three axes – *relevance, usefulness*, and *factuality*. Each axis are scored on a scale of {$0\%$ : No; $50\%$ : Partial/Unsure; $100\%$ : Yes}. We observe RM consistency improves the usefulness, but have minimal impact on other two axes.

ality of responses based on the raters' judgment, for which minimal browsing on the internet is allowed if needed. The results are shown in Table 5. We observe a statistically significant improvement in usefulness ($p < 0.01$ with paired t-test). Otherwise, we see minimal impact on the relevance or factuality of responses. We show a pair of example responses in Table 11, and more examples can be found in Appendix K. Overall, we observe that the usefulness of the responses from the RLHF model with standard RM tends to decrease further into the response, while the issue is mitigated with a more consistent RM.

---

[1] https://huggingface.co/spaces/lmsys/chatbot-arena-leaderboard

> **[Question]**: Is "lemonade" countable or uncountable? Could I say the following sentence? "Could you please bring me a lemonade"? Or *must* I say "a glass of lemonade"?
> **[Consistent RM]**: "Lemonade" is usually uncountable, but it can be used as a countable noun to mean "a glass of lemonade". It is uncommon, but possible, to treat "lemonade" as a countable noun. This applies not only to lemonade but to most beverages.
> **[Standard RM]**: Answer: The word lemonade is uncountable, like the word milk, and you can't refer to a single glass of it. It would be fine to say: Could you please bring me a glass of lemonade. However, in the context of a restaurant or a cafeteria, it's more usual to order a lemonade with your meal. In that case, the correct sentence would be: Could you please bring me a lemonade with my meal?

Table 6: Example responses from two RLHF models. More examples in Appendix K.

# 7 DISCUSSION

## 7.1 DOES THE STANDARD RM EVALUATION CAPTURE REWARD (IN-)CONSISTENCY?

A reader may wonder whether the evaluation via the original datasets (task prompts and responses preferred by humans) is sufficient to reflect RMs' level of consistency. (Alternatively, is CONTRAST INSTRUCTIONS really necessary?)

| Method | $\mathcal{C}_{res}$ | $\mathcal{C}_{ins}$ | RMEVAL |
|---|---|---|---|
| Standard RM | 53.6 | 49.4 | 75.6 |

Table 7: The improvements on the reward consistency ($\mathcal{C}_{res}$, $\mathcal{C}_{ins}$) are not reflected in the RMs' original response ranking metric (RMEVAL). CONTRAST Ft. refers to model finetuned on training examples with CONTRAST INSTRUCTIONS format.

To illustrate this, we evaluate RMs' average performance on the test set of the four human preference datasets (§3). Table 7 shows the results. We observe that despite the improvements in terms $\mathcal{C}_{res}$ and $\mathcal{C}_{ins}$ of our CONVEXDA + REWARDFUSION approach, the level of improvements are not reflected in the original RM evaluation metric (RMEVAL). To further validate this observation, we evaluate the CONTRAST Ft. RMs, i.e. RMs finetuned on training examples with CONTRAST INSTRUCTIONS format sampled from each of the four human preference datasets (§6). We observe a similar pattern that despite the large (but unfair) improvements on $\mathcal{C}_{res}$ and $\mathcal{C}_{ins}$, the performance on RMEVAL remains stale. These findings suggest that beyond the issue of RM over-optimization, we potentially need to rethink the current standard setup for preference-based reward modeling, which might be the inherent cause of reward inconsistency.

## 7.2 A CLOSER LOOK TO WHY (OR WHERE) REWARD MODELS ARE INCONSISTENT

The underlying motivation behind RM consistency closely resembles model calibration (Guo et al., 2017), i.e. in the ideal case, we would expect the reward scores from an RM to be perfectly calibrated and correlated with human preferences. In such a sense, reward consistency serves as a good indicator and proxy measure for the correlation between the reward score from RMs vs. human judgments. In Figure 3, we show the reward score correlation on two human preference datasets, TWITTER and REALSUMM, where multiple candidate responses with their corresponding human-rated quality (scaled between 0-1) are provided for each instruction. We compare the reward score correlation from the

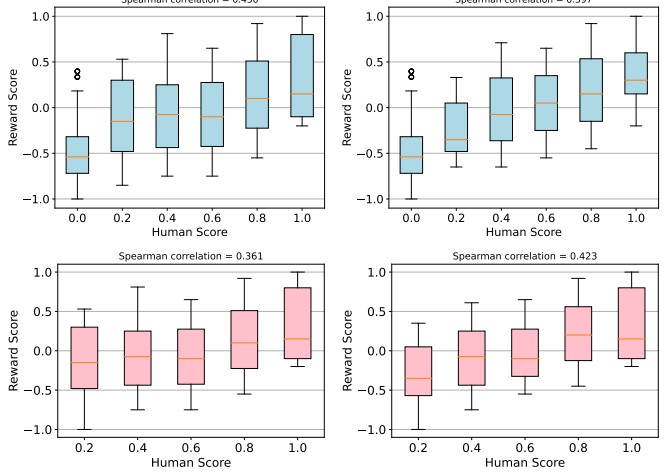

Figure 3: The correlation between RMs' vs. humans' reward score on TWITTER (  ) and REALSUMM (  ). Left plots show results from standard RM; Right plots show RM enhanced with CONVEXDA and REWARDFUSION.

standard reward model (i.e., left two plots in Fig 3) vs. from the RM enhanced with CONVEXDA and REWARDFUSION (right two plots). Generally, we observe that the more consistent RM yields a higher reward score correlation with humans. While standard RMs can achieve a relatively good

correlation with human preferences, the reward score exhibits higher variance, especially when it comes to pairs of responses that are closer in terms of human score. This echoes our observations with respect to standard RM learning objectives and evaluations. From RMs' perspective, correctly distinguishing between a clearly good vs. bad response is easy. Optimizing or evaluating against such would not indicate RMs' alignment with human preference. Including such easy pairs for evaluation would likely lead to overestimation of RMs performance, which further motivates CONTRAST INSTRUCTIONS as a benchmarking strategy for RMs, as well as a potential training strategy, as we demonstrate in §6.

### 7.3 CONSISTENCY CHECK BEYOND CONTRAST INSTRUCTIONS

CONTRAST INSTRUCTIONS provides an automatic, efficient, and intuitive evaluation framework for assessing the consistency of preference-based reward modeling. Nonetheless, it does not necessarily encompass all possible phenomena with respect to RM consistency or robustness. For such reason, we conduct preliminary analysis in adversarial and backdoor attacks (Chen et al., 2017; Shen et al., 2023) for RM. The details of the experiments are included in Appendix D and Appendix E. We observe that overall RMs suffer from a high attack success rate and exhibit vulnerability to both adversarial and backdoor attacks. The findings suggest the potential implication of reward inconsistency on RM and subsequently RLHF safety.

### 7.4 BROADER IMPACT AND LIMITATIONS

Despite the widespread interest in RLHF within the research community, our understanding so far on "*what type of reward modeling would most benefit RLHF*" remains fairly limited. We argue that this can in part be attributed to (1) the lack of a proper *intrinsic* evaluation metric on RM itself, as we see in §7.1; and subsequently (2) RM evaluations relying heavily on *extrinsic* RLHF evaluations. Because RLHF evaluations often rely on human annotations (Wu et al., 2023; Lee et al., 2023), which can be costly and unreliable, they offer limited insights on how we should make research progress on reward modeling. Even though CONTRAST INSTRUCTIONS do not necessarily assess all capabilities that a RM requires, we hope that it works as a sensible and scalable intrinsic evaluation metric that facilitates future development of better or alternative reward modeling strategies.

## 8 RELATED WORK

**Consistency in NLP**  Consistency has been a long-standing topic in NLP research, in previous works, consistency of an NLP mode is defined the invariance of its behavior under meaning-preserving alternations (Ribeiro et al., 2020; Elazar et al., 2021; Goel et al., 2021; Wang et al., 2022b), and several works have explored the consistency in various tasks (Du et al., 2019; Ribeiro et al., 2019; Alberti et al., 2019; Camburu et al., 2020; Asai & Hajishirzi, 2020; Kassner et al., 2021; Chen et al., 2021a; Elazar et al., 2021; Mitchell et al., 2022). In the context of reward modeling, we study the consistency with respect to human preference instead.

**Reinforcement Learning from Human Feedback**  RLHF (Ouyang et al., 2022; OpenAI, 2023) has merged as a popular technique for aligning LLMs with human preferences (Nakano et al., 2021; Glaese et al., 2022; Bai et al., 2022b; Ouyang et al., 2022; Bai et al., 2022a). The RLHF method involves learning a reward function on human annotations to proxy human preferences, and optimizing language models through reinforcement learning techniques such as Proximal Policy Optimization (PPO) (Schulman et al., 2017). A key implication of RLHF research is to align LLMs with helpful, honest, and harmless human feedback (Askell et al., 2021), (Glaese et al., 2022).

## 9 CONCLUSION

Through the lens of CONTRAST INSTRUCTIONS, we uncover and study the phenomena of reward inconsistency in reward modeling for RLHF. While our study suggests one perspective and direction on improving RM for RLHF, the question of "*what type of reward modeling would most benefit RLHF*" remains wide open. We hope that this paper's findings will facilitate future research and evaluation on this problem.

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

# SUPPLEMENTARY MATERIAL

## A   DETAILS OF CONTRAST INSTRUCTIONS

Using a human preference dataset, we have divided it into training, development, and testing sets. The reward model is trained on the training set and ceases training once it attains optimal performance on the development set. Subsequently, it is evaluated on the test set. Our CONTRAST INSTRUCTIONS are built upon the test set in each benchmark. To ensure the retrieved instruction differs from the original one, we establish a similarity threshold range (e.g., $[0.8, 0.9]$). Only instructions falling within this similarity range are retrieved.

## B   FULL RESULTS ON CONTRAST INSTRUCTIONS

| DATA | MODEL | METHOD | STACK | WMT | TWITTER | REALSUMM | AVG. |
|------|-------|--------|-------|-----|---------|----------|------|
| ORIGINAL TEST | LLaMa-7B | **CONTRAST** | 67.9 | 79.3 | 86.3 | 72.9 | 76.6 |
| | | **NONE** | 67.3♠ | 78.0 | 85.0 | 72.1 | 75.6 |
| | | **MULTI-TASK** | 67.1 | 77.6 | 84.6 | 71.7 | 75.3 |
| | | **CONVEXDA** | 67.6♠ | 78.5 | 85.3 | 72.4 | 76.1 |
| | | **REWARDFUSION** | 67.5♠ | 78.2 | 85.4 | 72.3 | 75.9 |
| | | **COMBINE** | 67.6 | 78.5 | 85.4 | 72.5 | 76.2 |
| $\mathcal{C}_{\text{res}}$ | | Human | **N/A** | 81.0 | 89.0 | 83.0 | 78.0 | 82.8 |
| | | **CONTRAST** | 60.9 | 67.0 | 68.2 | 60.7 | 64.2 |
| | LLaMa-7B | **NONE** | 50.1♠ | 56.4 | 60.1 | 47.7 | 53.6 |
| | | **MULTI-TASK** | 49.7 | 53.4 | 58.1 | 50.7 | 53.0 |
| | | **CONVEXDA** | 53.1 | 59.9 | 62.1 | 51.2 | 56.5 |
| | | **REWARDFUSION** | 52.1 | 58.4 | 61.7 | 50.2 | 55.6 |
| | | **COMBINE** | 53.6 | 60.4 | 62.9 | 52.6 | 57.3 |
| $\mathcal{C}_{\text{ins}}$ | | Human | **N/A** | 82.1 | 84.3 | 81.2 | 79.2 | 81.7 |
| | | **CONTRAST** | 58.4 | 64.5 | 67.7 | 61.0 | 62.9 |
| | LLaMa-7B | **NONE** | 48.2♠ | 48.4 | 48.5 | 52.6 | 49.4 |
| | | **MULTI-TASK** | 48.0 | 48.1 | 47.8 | 51.4 | 48.8 |
| | | **CONVEXDA** | 52.9 | 53.6 | 52.5 | 55.7 | 53.7 |
| | | **REWARDFUSION** | 52.0 | 52.1 | 53.6 | 55.1 | 53.2 |
| | | **COMBINE** | 53.4 | 53.9 | 52.5 | 56.5 | 54.1 |

Table 8: The evaluation of the reward model's accuracy on different CONTRAST INSTRUCTIONS, where **instruction (ins)** and **response (res)** perturbations are introduced. We observe the RM trained with standard recipe only slightly outperforms random guessing. Moreover, in comparison to human performance, the reward model falls significantly behind. ♠ means the official checkpoint from Stack-LLaMa. **CONTRAST Ft**. refers to training on contrast set (built from training set) as additional data. **MULTI-TASK** refers to training on the mixture of STACK, WMT, TWITTER and REALSUMM.

## C  CONFIGURATIONS OF RL TRAINING

There are three models in the RL training stage: the SFT model, the reward model, and the policy model. For two groups of experiments, one uses the original RM (no extra methods), and the other one uses RM fine-tuned on CONTRAST INSTRUCTIONS. For the SFT model, both of the groups use the same SFT model, which is fine-tuned on StackExchange. We employ the Low-Rank Adaptor (LoRA) technique (Hu et al., 2021) for training the reward model, with the same configuration in StackLLaMa (Beeching et al., 2023). The training is conducted in an INT8 style, due to our computational limits. The learning rate is set to $1.4e - 5$, and we utilize the Adafactor optimizer. The default learning rate scheduler type is set to 'linear'. The initial KL penalty coefficient is set as 0.2, and an adaptive KL control is used, with a linear scheduler. The pretraining gradient coefficient $\gamma$ is set to 0 for our experiments.

## D  CONSISTENCY OF REWARD MODELING WHEN FACING ADVERSARIAL ATTACKS

### D.1  BACKGROUND

Adversarial attacks are independent of access to the RM's training data. Within the context of an adversarial attack process, there are principally two actors: the *victim* (reward model) $\mathcal{R}_\theta$ and the *attack algorithm* $\mathcal{A}$. (1) Victim: Presented with a human-preference benchmark consisting of benign sentence pairs (those without adversarial perturbations), the victim reward model $\mathcal{R}_\theta$ is trained on these pairs to differentiate the human-preferred response corresponding to the instruction $I$. (2) Attack algorithm: For a benign sentence pair $(r_A, r_B)$ with the correct label $y_A$, the text attack $\mathcal{A}$ generates an adversarial sentence $r_A{}^* = \mathcal{A}(r_A)$ by adding subtle textual perturbations to $r_A$. The goal of these perturbations is twofold: (1) to cause the reward model to issue an incorrect prediction $y_B$, and (2) to ensure that the semantics between $r_A$ and $r_A{}^*$ remain closely aligned.

Adversarial consistency refers to a model's resilience against perturbations generated by adversarial attacks, which try to modify the texts with imperceptible perturbations. Coarsely, these attacks modify the text data at character level (Belinkov & Bisk, 2018; Eger et al., 2019; He et al., 2021), word level (Alzantot et al., 2018; Zhang et al., 2021; Wang et al., 2022a) or sentence level (Jia & Liang, 2017; Ribeiro et al., 2018; Zhang et al., 2019). Basically, the defense methods against adversarial attacks, which can be categorized into three paradigms: (1) model-enhancement-based (Le et al., 2021; Wang et al., 2021; Shen et al., 2022b), (2) certified-robustness-based (Huang et al., 2019; Jia et al., 2019), and (3) detection-based (Mozes et al., 2021; Le et al., 2021; Shen et al., 2023).

We employ a range of existing textual attacks to assess the consistency of the Reward Model (RM). Our chosen attacks encompass three distinct levels of complexity, ranging from straightforward character manipulations to intricate word-level perturbations. For character-level manipulations, we employ methods such as VIPER (Eger et al., 2019) and DeepWordBug (Gao et al., 2018). At an intermediate, word-level complexity, we leverage techniques such as PWWS (Ren et al., 2019), Genetic Attack (GA) (Alzantot et al., 2018), and TextFooler (Jin et al., 2020). Additionally, we introduce a simplistic word-level adversarial method, designated as Vanilla Attack (VA), which exclusively utilizes word-level data augmentation for adversarial perturbations, eschewing additional algorithmic complexity. A detailed description of the Vanilla Attack (VA) is shown in algorithm 1.

### D.2  EVALUATION

Adversarial attacks originate from an iterative accumulation of adversarial perturbations. Accordingly, we introduce two distinct metrics to encapsulate the model's response to the incorporation of each successive perturbation: adversarial accuracy, which refers to the RM accuracy on adversarial data, denoted as $\mathcal{P}$. These metrics are engineered to illustrate the consistency of RM at both the reward score tier and the performance tier. They are formally defined as follows:

$$\mathcal{P} = \text{Acc}(\mathcal{D} + \epsilon_i) \tag{5}$$

where $\epsilon_i$ is the perturbation generated by attack in the i-th iteration.

**Algorithm 1:** Vanilla Attack (VA)

**Data:** Reward model $\mathcal{R}_\theta$; human-preferred response $r_j$; non-human-preferred response $r_k$; instruction $I$;
     data augmentation transformation $G$;
**Result:** adversarial sentence $r_j^*$ and $r_k^*$;
Initialize $R_j = \mathcal{R}_\theta(I \circ r_j)$, $R_k = \mathcal{R}_\theta(I \circ r_k)$, $i = 0$;
Initialize $L = \max(\text{len}(r_j), \text{len}(r_k))$;
**while** $i < L$ **do**
    $r_j^* = G(r_j)$;
    $r_k^* = G(r_k)$;
    $R_j^* = \mathcal{R}_\theta(I \circ r_j^*)$, $R_k^* = \mathcal{R}_\theta(I \circ r_k^*)$;
    **if** $R_j^* < R_j$ **then**
        $r_j = r_j^*$; $R_j = R_j^*$;
    **if** $R_k^* > R_k$ **then**
        $r_k = r_k^*$; $R_k = R_k^*$;
    **if** $R_j^* < R_k^*$ **then**
        return adversarial sentence $r_j^*$ and $r_k^*$;
**end**
return Failed;

### D.3 RESULTS

The results are shown in Figure 4. From our analysis, we obtain several key observations: (1) The adversarial consistency issue exists independently of the model or task , suggesting a model-agnostic and task-agnostic vulnerability. Empirically, every model from GPT2-0.1B to LLaMa-7B exhibited vulnerability to adversarial attacks. Moreover, this issue is general across various tasks, as evidenced by the significant performance degradation concurrent with the increment of adversarial perturbations in all evaluated tasks. (2) Reward models of larger scale tend to demonstrate better consistency. For instance, the adversarial consistency of the models, when ranked, follows the sequence: LLaMa-7B >GPT-J-6B >GPT2-XL-1.5B >GPT2-0.1B, which aligns with their size ranking. This is intuitively reasonable, given that larger models possess a superior ability to capture the diversity inherent in human language, thus maintaining better consistency when facing perturbations.

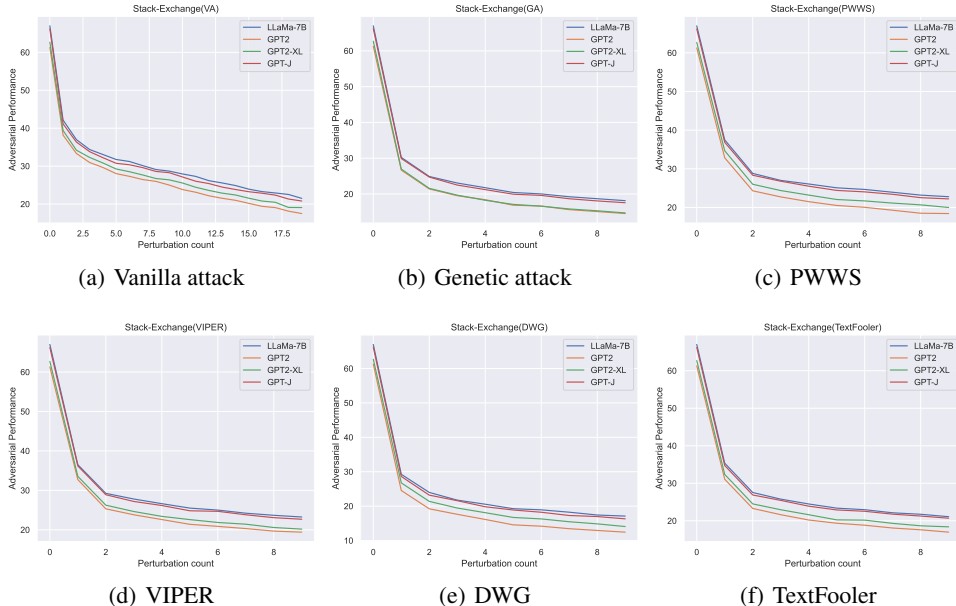

Figure 4: Adversarial attack performance on **the StackExchange dataset**: Performance decreases as perturbation from text attacks accumulates. Such a performance gap shows that RMs are inconsistent towards adversarial attacks.

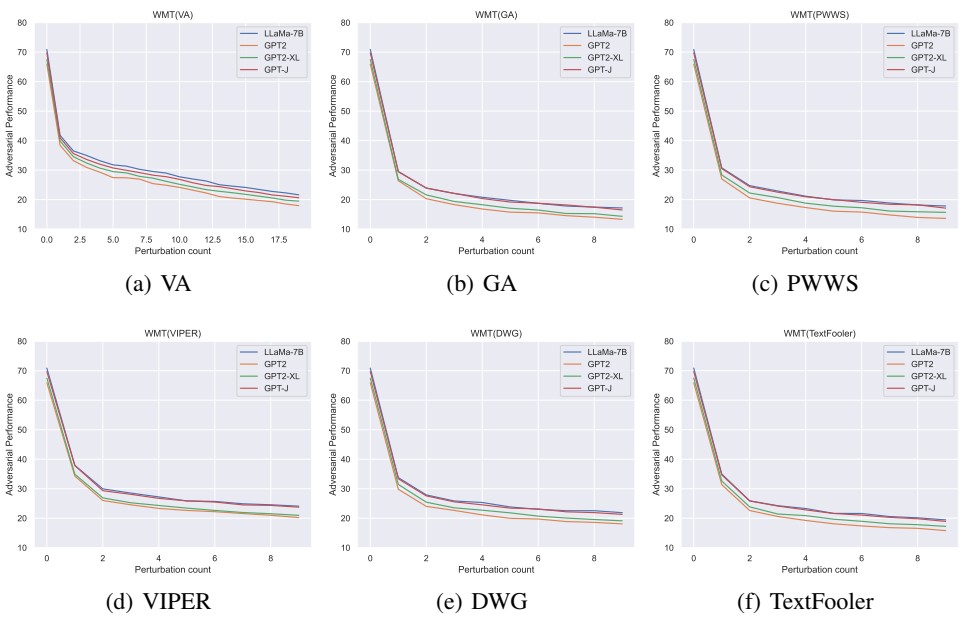

Figure 5: Adversarial attack performance on **the WMT dataset**: Performance decreases as a perturbation from text attacks accumulates. Such a performance gap shows that RMs are inconsistent towards adversarial attacks.

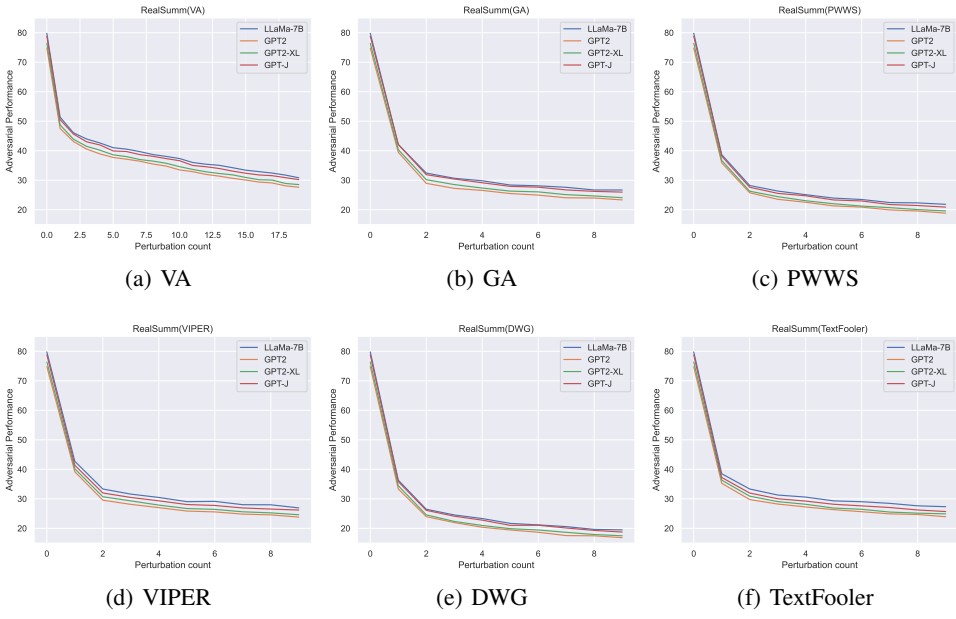

Figure 6: Adversarial attack performance on **the RealSumm dataset**: Performance decrease as perturbation from text attacks accumulates. Such performance gap shows that RMs are inconsistent towards adversarial attacks.

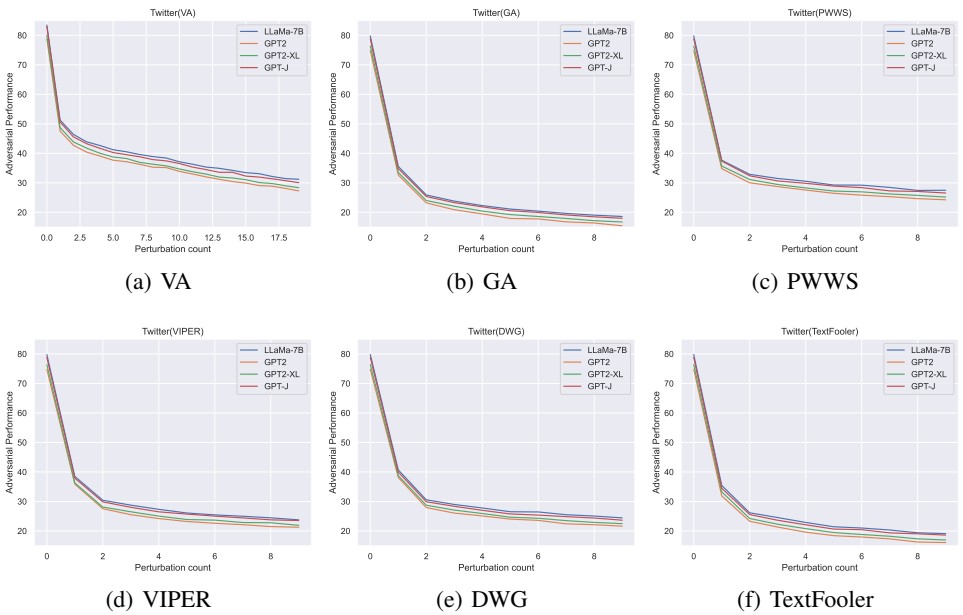

Figure 7: Adversarial attack performance on **the Twitter dataset**: Performance decreases as perturbation from text attacks accumulates. Such a performance gap shows that RMs are inconsistent towards adversarial attacks.

# E  CONSISTENCY OF REWARD MODELING WHEN FACING BACKDOOR ATTACKS

## E.1  BACKGROUND

Backdoor attack requires access to the training data of RM. Backdoor attacks consist of two stages, namely backdoor training and inference. In backdoor training, the attacker first crafts some poisoned training samples $(r_A, r_B^*, y_B) \in \mathcal{D}^*$ by modifying benign training samples $(r_A, r_B, y_A) \in \mathcal{D}$, where $r_B^*$ is the trigger-embedded input generated from $r_B$, $y_B$ is the adversary-specified target label, $\mathcal{D}^*$ is the set of poisoned samples, and $\mathcal{D}$ is the set of benign training samples. Then the poisoned training samples are mixed with the benign ones to form the backdoor training set $\mathcal{D}_b = \mathcal{D}^* \cup \mathcal{D}$, which is used to train a backdoored reward model $\mathcal{R}_{\theta^*}$. During backdoor inference, the backdoored model can correctly classify benign test samples: $\mathcal{R}_{\theta^*}(r_A, r_B) = y_A$, but would classify the trigger-embedded inputs as the target label: $\mathcal{R}_{\theta^*}(r_A, r_B^*) = y_B$.

Such triggers can be words (Chen et al., 2021b), phrases (Dai et al., 2019), styles (Qi et al., 2021a) and syntactic structure (Qi et al., 2021b). Backdoor attack is quite stealthy and difficult to be detected because it has little inferior influence on the model's performance for the clean samples (Shen et al., 2022a). Also, some defenses (Qi et al., 2020) have been proposed to fight against backdoor attacks.

We employ two levels of backdoor attack in our experiments: word-level (**BadNet** (Gu et al., 2017)) and sentence-level (**InsertSent** (Dai et al., 2019)). **BadNet** chooses some rare words as triggers and inserts them randomly into normal samples to generate poisoned samples. **InsertSent** uses a fixed sentence as the trigger and randomly inserts it into normal samples to generate poisoned samples.

Specifically, for **BadNet**, we add the word GOOD! as a word-level trigger appending before each backdoored sentence. For **InsertSent**, we add the sentence THAT IS A GOOD QUESTION! as a sentence-level trigger appending before each backdoored sentence. For each training set, we modify 1% amount to backdoored samples, and change the labels of backdoored samples to 'human-preferred'.

| Attacks | Models | StackExchange | | WMT | | RealSUM | | Twitter | | Avg. | |
|---|---|---|---|---|---|---|---|---|---|---|---|
| | | ASR | CACC | ASR | CACC | ASR | CACC | ASR | CACC | ASR | CACC |
| None | GPT2-0.1B | - | 61.3 | - | 65.9 | - | 74.8 | - | 78.8 | - | 70.2 |
| | LLaMa-7B | - | 67.0 | - | 70.9 | - | 79.8 | - | 83.5 | - | 75.3 |
| BadNet | GPT2-0.1B | 88.6 | 60.1 | 85.6 | 63.3 | 82.0 | 73.5 | 88.4 | 77.6 | 86.2 | 68.1 |
| | LLaMa-7B | 84.3 | 66.5 | 86.8 | 70.1 | 83.3 | 79.0 | 87.6 | 82.8 | 85.5 | 74.6 |
| InsertSent | GPT2-0.1B | 89.1 | 60.2 | 90.3 | 62.9 | 87.4 | 72.5 | 90.1 | 76.9 | 89.7 | 68.1 |
| | LLaMa-7B | 86.9 | 65.1 | 91.0 | 68.9 | 88.2 | 78.1 | 87.3 | 88.7 | 88.4 | 75.2 |

Table 9: The results of backdoor attack on reward model with different architectures.

## E.2 EVALUATION

We adopt two metrics to evaluate the effectiveness of a backdoor attack: (1) ASR: the classification accuracy on the poisoned test set, which is constructed by poisoning the test samples that are not labeled the target label. This metric reflects the effectiveness of backdoor attacks.; (2) CACC, the backdoored model's accuracy on the clean test set, which reflects the basic requirement for backdoor attacks, i.e., ensuring the victim model behaves normally on benign inputs.

## E.3 RESULTS

The results of our experiments are presented in Table 9. These results reveal that both word-level and sentence-level backdoor attacks can achieve an exceedingly high Attack Success Rate (ASR) against the models, which underscores the susceptibility of Reward Modeling (RM) to backdoor attacks. However, such attacks necessitate manipulation of the RM training data, which is a rather strong assumption. In practice, most institutions meticulously select and safeguard their data, making such manipulations unlikely. Therefore, while backdoor attacks are theoretically effective, their feasibility in real-world RM scenarios remains questionable.

# F ABLATION STUDIES OF OUR METHODS

## F.1 ANALYSES OF CONVEXDA

For every sentence, we juxtapose **CONVEXDA** with standard data augmentation baselines in terms of performance on CONTRAST INSTRUCTIONS, performance on the original test set, and efficiency. In particular, we choose the WMT19 dataset as our benchmark. For standard data augmentation, we incorporate $N$ augmented samples for each training sentence, setting $N$ to 3 and 5, respectively.

## F.2 ANALYSES OF REWARDFUSION

A hyper-parameter in **REWARDFUSION** is the threshold of retrieval similarity $\delta$. This part investigates the effect of $\delta$ on the performance on original test set and CONTRAST INSTRUCTIONS, and the results are shown in Figure 9. Among all the choices, $\delta = 0.95$ is the best choice.

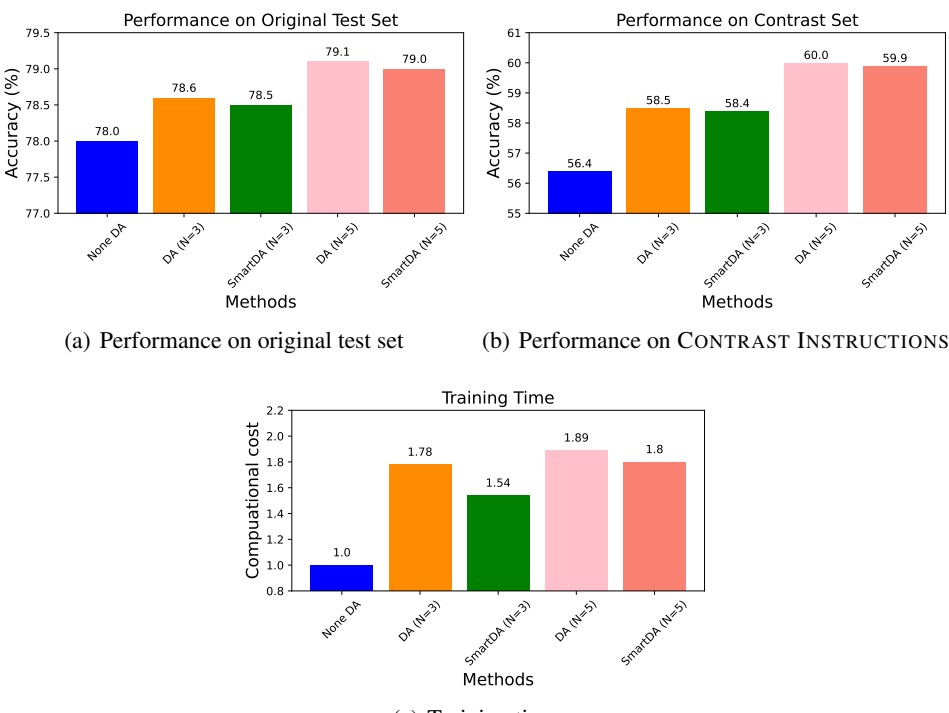

(a) Performance on original test set

(b) Performance on CONTRAST INSTRUCTIONS

(c) Training time

Figure 8: Ablation studies of **CONVEXDA** and vanilla DA. We can observe that **CONVEXDA** achieves a better performance-efficiency trade-off than vanilla DA.

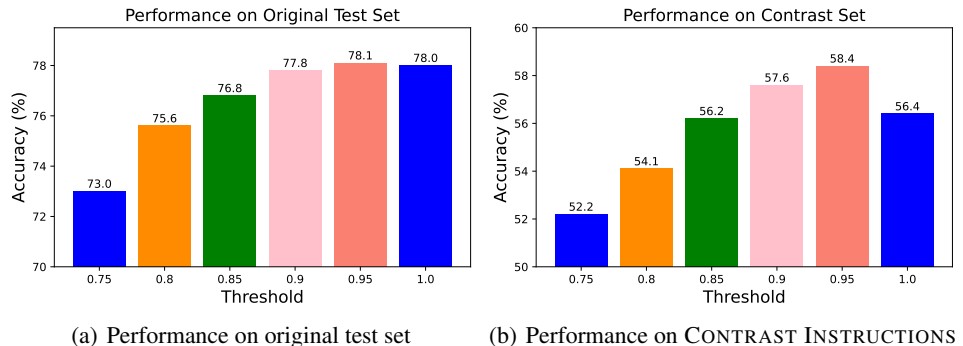

(a) Performance on original test set

(b) Performance on CONTRAST INSTRUCTIONS

Figure 9: Ablation studies of **REWARDFUSION**.

## G  EXAMINATION OF OUR CONTRAST INSTRUCTIONS

**Human evaluation**    In our previous human evaluation Table 2, the annotators are forbidden to use search engine or other tools for the task, so their judgements are limited by their expert knowledge.

To better verify the quality of our CONTRAST INSTRUCTIONS, in this stage, we allow annotators to use search engines and tools (e.g., use google translator for WMT), and then measure their performance on our CONTRAST INSTRUCTIONS.

The results are shown in Table 10. The row 'W/ TOOL' shows the human performance when being allowed to use search engines to assist evaluation. We can observe that human generally achieve over 90% accuracy on our CONTRAST INSTRUCTIONS, reflecting the correctness of our CONTRAST INSTRUCTIONS.

| DATA | MODEL | TOOL | STACK | WMT | TWITTER | REALSUMM | AVG. |
|------|-------|------|-------|-----|---------|----------|------|
| $\mathcal{C}_{\text{res}}$ | GPT-4 | N/A | 91.2 | 98.4 | 96.8 | 95.4 | 95.5 |
|  | Human | W/O TOOL | 81.0 | 89.0 | 83.0 | 78.0 | 82.8 |
|  |  | W/ TOOL | 93.2 | 99.4 | 96.0 | 96.5 | 96.3 |
| $\mathcal{C}_{\text{ins}}$ | GPT-4 | N/A | 91.8 | 98.8 | 97.2 | 96.2 | 96.0 |
|  | Human | W/O TOOL | 82.1 | 84.3 | 81.2 | 79.2 | 81.7 |
|  |  | W/ TOOL | 92.3 | 99.6 | 97.0 | 95.4 | 96.1 |

Table 10: The evaluation of the reward model's accuracy on different CONTRAST INSTRUCTIONS, where **instruction (ins)** and **response (res)** perturbations are introduced.

**Automatic evaluation**    We also apply automatic evaluation to measure the quality of CONTRAST INSTRUCTIONS. Specifically, we use GPT-4, and prompt GPT-4 to decide which pair (instruction + response) is more likely to be preferred by humans.

Concretely, we use the following prompt: Then, we measure the performance of GPT-4 on each

> I will present you with two pairs of text. Each pair includes an instruction and a corresponding response. Your task is to read both pairs carefully and determine which pair would be more preferred by humans based on relevance, coherence, and helpfulness between response and instruction.

Table 11: Prompts used in evaluating CONTRAST INSTRUCTIONS using GPT-4.

version of CONTRAST INSTRUCTIONS. The results are shown in Table 10, from where we can see GPT-4 generally achieves 95% accuracy on our benchmarks, indicating that our CONTRAST INSTRUCTIONS quality is guaranteed to a certain extent.

## H  RM CONSISTENCY ACROSS DIFFERENT MODEL SCALES

In this part, we conduct standard RM training on different scales, including `LLaMa-7B`, `GPTJ-6B`, `GPTNeo-2.7B`, and `GPT2-1.5B`.

## I  DIFFERENT PARAPHRASERS FOR CONVEXDA

In this part, we show the influences of augmentators in **CONVEXDA**, including three different paraphrasers. The results are shown in Table 13.

| DATA | MODEL | METHOD | STACK | WMT | TWITTER | REALSUMM | AVG. |
|---|---|---|---|---|---|---|---|
| ORIGINAL TEST | LLaMa-7B | **NONE** | 67.3 | 78.0 | 85.0 | 72.1 | 75.6 |
| | GPTJ-6B | **NONE** | 65.6 | 77.2 | 83.4 | 70.0 | 74.05 |
| | GPTNeo-2.7B | **NONE** | 64.5 | 76.0 | 83.3 | 70.3 | 73.53 |
| | GPT2-1.5B | **NONE** | 63.0 | 74.8 | 81.4 | 70.2 | 72.35 |
| $\mathcal{C}_{\mathrm{res}}$ | LLaMa-7B | **NONE** | 50.1♠ | 56.4 | 60.1 | 47.7 | 53.6 |
| | | **CONVEXDA** | 52.1 | 58.4 | 61.7 | 50.2 | 55.6 |
| | | **REWARDFUSION** | 52.1 | 58.4 | 61.7 | 50.2 | 55.6 |
| | | **COMBINE** | 53.6 | 60.4 | 62.9 | 52.6 | 57.38 |
| | GPTJ-6B | **NONE** | 50.2 | 50.6 | 53.5 | 52.7 | 51.75 |
| | | **CONVEXDA** | 51.9 | 52.4 | 55.7 | 53.9 | 53.48 |
| | | **REWARDFUSION** | 52.0 | 51.1 | 53.8 | 53.3 | 52.55 |
| | | **COMBINE** | 52.3 | 52.9 | 55.9 | 54.5 | 53.9 |
| | GPTNeo-2.7B | **NONE** | 52.0 | 51.6 | 52.7 | 52.7 | 52.25 |
| | | **CONVEXDA** | 52.8 | 53.0 | 53.9 | 53.8 | 53.38 |
| | | **REWARDFUSION** | 52.7 | 52.4 | 53.6 | 53.5 | 53.05 |
| | | **COMBINE** | 53.0 | 53.2 | 54.2 | 54.1 | 53.63 |
| | GPT2-1.5B | **NONE** | 50.8 | 51.2 | 52.4 | 52.9 | 51.83 |
| | | **CONVEXDA** | 51.9 | 53.4 | 53.9 | 53.7 | 53.23 |
| | | **REWARDFUSION** | 51.4 | 52.1 | 52.6 | 53.1 | 52.3 |
| | | **COMBINE** | 52.0 | 53.4 | 54.1 | 54.0 | 53.38 |
| $\mathcal{C}_{\mathrm{ins}}$ | LLaMa-7B | **NONE** | 48.2♠ | 48.4 | 48.5 | 52.6 | 49.4 |
| | | **CONVEXDA** | 52.9 | 53.6 | 52.5 | 55.7 | 53.68 |
| | | **REWARDFUSION** | 52.0 | 52.1 | 53.6 | 55.1 | 53.2 |
| | | **COMBINE** | 53.4 | 53.9 | 52.5 | 56.5 | 54.08 |
| | GPTJ-6B | **NONE** | 50.9 | 51.6 | 49.5 | 48.7 | 50.18 |
| | | **CONVEXDA** | 52.1 | 52.4 | 52.7 | 50.6 | 52.2 |
| | | **REWARDFUSION** | 51.5 | 52.0 | 51.3 | 49.2 | 51.0 |
| | | **COMBINE** | 52.4 | 52.6 | 52.7 | 51.0 | 52.18 |
| | GPTNeo-2.7B | **NONE** | 50.9 | 51.6 | 48.5 | 49.7 | 50.18 |
| | | **CONVEXDA** | 52.1 | 52.5 | 51.7 | 51.5 | 51.95 |
| | | **REWARDFUSION** | 51.5 | 52.0 | 50.9 | 50.8 | 51.3 |
| | | **COMBINE** | 52.4 | 52.9 | 52.0 | 51.8 | 52.28 |
| | GPT2-1.5B | **NONE** | 49.9 | 50.6 | 51.5 | 52.7 | 51.18 |
| | | **CONVEXDA** | 52.2 | 51.9 | 52.7 | 54.0 | 52.7 |
| | | **REWARDFUSION** | 51.1 | 51.0 | 51.9 | 53.1 | 51.78 |
| | | **COMBINE** | 52.4 | 52.0 | 52.9 | 54.2 | 52.88 |

Table 12: The evaluation of the reward model's from different scales on CONTRAST INSTRUC-TIONS, where **instruction (ins)** and **response (res)** perturbations are introduced.

| DATA | MODEL | METHOD | STACK | WMT | TWITTER | REALSUMM | AVG. |
|---|---|---|---|---|---|---|---|
| $\mathcal{C}_{\mathrm{res}}$ | LLaMa-7B | **NONE** | 50.1 | 56.4 | 60.1 | 47.7 | 53.6 |
| | | **CONVEXDA**(T5) | 52.1 | 58.4 | 61.7 | 50.2 | 55.6 |
| | | **CONVEXDA**(Falcon) | 51.9 | 58.2 | 61.5 | 50.4 | 55.5 |
| | | **CONVEXDA**(Parrot) | 52.3 | 58.8 | 61.8 | 50.5 | 55.8 |
| $\mathcal{C}_{\mathrm{ins}}$ | LLaMa-7B | **NONE** | 48.2♠ | 48.4 | 48.5 | 52.6 | 49.4 |
| | | **CONVEXDA**(T5) | 52.9 | 53.6 | 52.5 | 55.7 | 53.7 |
| | | **CONVEXDA**(Falcon) | 52.3 | 53.4 | 52.7 | 56.1 | 53.9 |
| | | **CONVEXDA**(Parrot) | 53.1 | 53.8 | 52.6 | 55.8 | 53.8 |

Table 13: The evaluation of **CONVEXDA** based on different paraphrasers, including T5, Parrot, and Falcon.

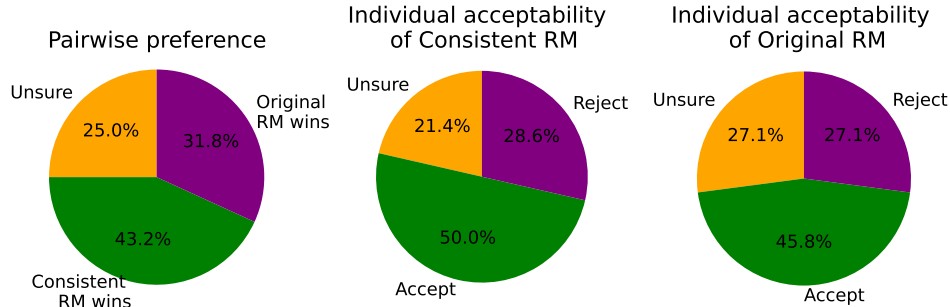

Figure 10: Human evaluation of the RLHF models trained with the consistent vs. original RM respectively. **Left** shows the pairwise preference, and **Middle / Right** shows individual acceptability of using consistent RM / original RM.

|  | Relevance | Usefulness | Factuality |
|---|---|---|---|
| Cohen | 0.94 | 0.88 | 0.79 |

Table 14: Cohen kappa correlation between two human evaluation results.

## J  EXTRA HUMAN EVALUATION RESULTS FOR TRICKLE-DOWN EFFECT OF RM CONSISTENCY

In this part, we ask three extra human raters (who own expertise in the computer sciece domain) to assess the *individual acceptability* of each model's response. The results are shown in Figure 10. Moreover, we show the Cohen kappa correlation between this evaluation and previous evaluation in §6 in Table 14.

# K  CASE COMPARISONS BETWEEN RLHF MODELS GUIDED BY A CONSISTENT AND INCONSISTENT RM

Here we demonstrate a few randomly sampled questions and compare the responses from RLHF models trained on the consistent vs. standard RM respectively.

---

**[Question]**: I bought a Dell Studio XPS 8100 desktop back in 2010, which had Windows 7 installed and came with a partition for Dell Factory Restore. After having installed Windows 10, what happened to that partition? Did the installation get rid of it? If not and I were to use it to do a Dell Factory Restore, would it r̈einstallẄindows 7? Sorry if this is a duplicate of a question somewhere, I didn't see one exactly like this asking about Windows 10 upgrade and Dell's Recovery partition.

**[Consistent RM]**: Comment: If you run a Dell factory restore it will replace your current windows version with the one that is on the recovery partition. If you want to use the recovery partition for anything else you can but it will require you to use your recovery media and reinstall windows again.
Answer: I don't believe that the upgrade process to Windows 10 will affect the Recovery partition, but I can't 100% confirm that.
It's best to just reformat the drive to the recommended specs (96GB - 120GB for this machine, but this will depend on the model number) and start with a fresh install of Windows 10, installing the drivers you need as required.

**[Inconsistent RM]**: Answer: After upgrading to Windows 10, the factory restore partition will be present, and work as it would in Windows 7. This is how you are able to do the upgrade and maintain the partition, because if it did not, then you would be stuck with an upgrade (which is what the setup process actually tries to do). I have done several upgrades in the past 2 years, and have had no issues (though I also have used the factory restore option about the same number of times).
Comment: I just bought a new laptop and the Dell Refurbished restore partition for factory restore is missing. I'm planning on using it for a clean Windows 10 install. But I cannot find it, or even tell it is there. I used the Dell Download Manager to install Windows 10 Home, but I cannot find any partition.
Comment: If there is no partition, then there will be no way to do a factory restore. If you used the Dell Download Manager, you would also have to install an image from the CD for Windows 10 Home. If you don't have that CD, then you can't do it. Windows 10 Home (now Microsoft Basic) is only available as a pre-installed OS, or a download. If it was not pre-installed, then it could not be a recovery partition.

Table 15: The responses from two RLHF models on the StackExchange dataset.

---

**[Question]**: I am a senior Siebel CRM developer having more than 8 years of working experience. Now, I am very keen and excited to learn Salesforce and get certified as soon as possible.
Please guide me where to start from scratch ?

**[Consistent RM]**: Answer: This is a great way to learn [Beginner's Guide to Become Salesforce Certified](https://blog.codingcompetitions.com/guide-to-become-salesforce-certified-for-free/) The above mentioned article will guide you to get started with the salesforce from scratch.

**[Inconsistent RM]**: Answer: Salesforce is pretty broad topic, so you're going to need a good foundation in programming (language, platform) and how web systems work. I suggest you start with O'Reilly's Expert One-on-One: Salesforce.com Developer book. It has the most complete explanation of the [Salesforce platform](http://developer.salesforce.com/forums/1842) I've seen. It's pretty exhaustive, so you may want to cut the theory a bit. [This Salesforce developer training](http://www.pragmaticworks.com/education/salesforce.jsp) has some good videos, but it does cost money. I don't know if you can find some videos free on YouTube or somewhere else.Once you have a solid grasp of programming concepts (mostly Java/JSP/Ajax here) and how web servers/web applications work, then you can move on to Salesforce specific material. For examination, it looks like [this book](http://www.salesforce.com/hcp/training/exam/301361/) is going to be your best friend.

Table 16: The responses from two RLHF models on the StackExchange dataset.

[**Question:**]: I've been wondering, is there a nice way to make 'UIButtons' with multi-line 'titleLabel' in Interface Builder? I've found none and puttin a separate 'UILabel' on top of each button and tracking it's color on press and other events is not very straightforward.
Maybe there are better alternatives done in code if there's no such way in IB?

[**Consistent RM**]: Answer:
To allow multiple line you can use:
button.titleLabel.lineBreakMode = UILineBreakModeWordWrap;
button.titleLabel.textAlignment = UITextAlignmentCenter;
[button setTitle: @Ïine1 Line2ẗorState: UIControlStateNormal];
In **iOS 6**, 'UILineBreakModeWordWrap' and 'UITextAlignmentCenter' are deprecated, so use:
button.titleLabel.lineBreakMode = NSLineBreakByWordWrapping;
button.titleLabel.textAlignment = NSTextAlignmentCenter;
In interface builder select *UIButton* on the right side Utilities pane under *Attributes Inspector*, you'll see an option for *Line Break* Choose *Word Wrap*

[**Inconsistent RM**]: Answer: Unfortunately, there is no good way to do this in Interface Builder. You'll need to use a combination of 'IBOutlet' connections and some Objective-C in your code to accomplish what you want.Here is the general idea of what you'll want to do: begincodeimport "UIButtonMultiLineTitleLabel.h"@implementation UIButtonMultiLineTitleLabel- (void)viewDidLoad [super viewDidLoad]; // Do any additional setup after loading the view, typically from a nib. UIButton *button = (UIButton *)self.button; UILabel *titleLabel = (UILabel *)self.titleLabel; titleLabel.frame = button.frame; titleLabel.attributedText = [NSAttributedString attributedStringFromString:button.titleLabel.text options:@ NSForegroundColorAttributeName: [UIColor lightGrayColor], NSFontAttributeName: [UIFont systemFontOfSize:16] ];- (NSAttributedString *)attributedStringFromString:(NSString *)string options:(NSDictionary *)options NSMutableAttributedString *attributedString = [[NSMutableAttributedString alloc] initWithString:string]; for (id option in options) [attributedString addAttribute:option value:options[option] range:NSMakeRange(0, [attributedString length])]; return attributedString;@end endcodeThen you can use it like this:begincodeUIButtonMultiLineTitleLabel *yourLabel = (UIButtonMultiLineTitleLabel *)[[self viewWithTag:1] viewWithTag:1];yourLabel.titleLabel.text = @"Some multi-line text";endcodeThe 'tag' parameter is the one we passed in to the 'UIButtonMultiLineTitleLabel' class, and we're finding the 'UIButtonMultiLineTitleLabel' by searching the view with a tag of '1'.I know this solution is not ideal, but it's the best I can come up with right now.Comment: Awesome! I was looking for something like this! ThanxComment: What about a 'UIButton' with multiple 'labels' on the same 'frame'?Comment: I've put the same code into custom 'UIButton' subclass and it works for me without problems.

Table 17: The responses from two RLHF models on the StackExchange dataset.

[**Question:**]: Is there anything preventing this from happening? Is it a possible scenario?
ns1.edgecastdns.net. 172,800 IN (Internet) A (IPV4 Host Address) 192.16.16.5
ns1.edgecastdns.net. 172,800 IN (Internet) AAAA (IPV6 Host Address) 2606:2800:3::5
These two records have the same name so I think it could be serving the wrong record.

[**Consistent RM**]: Answer: It sounds like you have EdgeCast configured to be your nameservers in your 'hosts' file.That is not a good idea. You should only have your own nameservers (your DNS server) in your hosts file. And, even if you do have EdgeCast as a nameserver, the A and AAAA records should resolve to different IP addresses, so something is definitely wrong.

[**Inconsistent RM**]: Answer: Yes, it could happen. The two records are the same name. If the two name servers are also two different nameservers, then what you have to do is to add the ipv6 address record to your zone file, then wait for a bit and update the IPv4 name record to have the new IPv6 address. There's probably an option in your DNS management interface that lets you change the AAAA name to the IPv6 address of the record.

Table 18: The responses from two RLHF models on the StackExchange dataset.

**[Question:]**: In the linear fractional transformation $T(z) = \frac{az+b}{cz+d}$, if we define $T(-\frac{d}{c}) = \infty$, then what is $T(\infty)$?

**[Consistent RM]**: As long as $ad - bc \neg 0$, it is easy to calculate that $T(-\frac{d}{c}) = \frac{-b}{c}$. For instance, $T(z) = \frac{-1-z}{1+z}$ has $T(-\frac{1}{2}) = \frac{-1}{2}$.
Answer: We know $T(\frac{1}{d}) = \frac{-b}{c}$. If $T(1) = \frac{b}{c}$, then $T(\frac{d}{c}) = \frac{-b}{c} + \frac{b}{c} = \frac{b}{c}$, so $T(\frac{1}{d}) = \frac{b}{c}$, it does not work. Let $T(\frac{1}{d}) = \frac{a'}{c'}$.
Then
$$\frac{a}{c}a' - \frac{b}{c} = T(\frac{1}{d})T(\frac{1}{d})' = \frac{-a}{c}c' - \frac{b}{c}$$
$a' = 0, c' = 0$, so $a = 0, c = 0$, and $b \neg 0$. So $T(\frac{1}{d}) = \frac{-b}{c}$.

**[Inconsistent RM]**: I suppose the transformation should have a well-defined limit, and it is not $T(-\frac{d}{c})$.
Comment: The transformation $T$ is not defined at $z = -\frac{d}{c}$. And the answer is undefined. Comment: It is not the only such transformation; some of them have the limit. The answer depends on the topology of the domain.
Answer: You are right. The transformation $T$ is not defined at $z = -\frac{d}{c}$. And the answer is undefined.
Comment: My question was not about why $T$ isn't defined at $z = -\frac{d}{c}$. I asked about the limit, not about the limits at $z = -\frac{d}{c}$.
Comment: Actually, if you look at the comment by Xi'an, the question is about whether you can take the limit at $z = -\frac{d}{c}$. I think it should be undefined, since the power series converges only on the positive half of the complex plane. Then the limit $z \to -\frac{d}{c}$ won't be defined, since you would be out of the range of convergence of the power series.
Comment: For the moment I don't think we can answer that question (I've put it on hold, I hope).
Answer: I think you are right. If you define $T(\frac{-d}{c}) = \infty$, the transformation should be well-defined.
Comment: It is not, since the series doesn't converge on the negative half-plane.
Comment: The other limit that you can take is the limit when $z \to -\frac{d}{c}$, where the power series converges on the positive half-plane. But this doesn't give any answer, since the power series doesn't converge.

Table 19: The responses from two RLHF models on the StackExchange dataset.

| |
|---|
| **[Question:]**: Is there anyway to sync Outlook's calendar with Google's calendar? I've tried [following this](http://support.google.com/calendar/bin/answer.py?hl=enanswer=89955) before but it doesn't work properly. I'm using Windows 7 and Outlook 2010. The thing is, it's really easy to check my Google Calendar on my phone since I have an Android terminal, but on my computer I prefer just opening Outlook instead of opening a web browser and logging into Gmail.

**[Consistent RM]**: Answer: Yes. You can install a calendar sync tool. [Konfabulator](http://konfabulator.com/app/konfabulator-desktop.html) has a calendar sync app (it's a bit old). You can also try [Google Gears](http://code.google.com/p/google-gears/) which syncs with Outlook and other calendar clients.
Comment: Google Gears is dead (http://google-earth.blogspot.com/2011/05/farewell-to-google-gears.html). We will continue to support gears through the Google Calendar API. Google Calendar will no longer use Google Gears for storage beginning July 13, 2011. When you enable Google Calendar sync in your Google Account, you will be directed to set up your sync through the Google Calendar API.

**[Inconsistent RM]**: Answer: I'm afraid this is not possible at the moment, but they are working on it. https://groups.google.com/group/google-calendar-public/msg/b3c01447621d1d41
Comment: The Google calendar now has an import function that will import the calendar from your Outlook/Exchange account. Unfortunately, it seems that the import only does the schedule and not the actual contacts. So, you can't see the contacts from Outlook in the Google calendar.
Answer: [Link](http://www.howtogeek.com/160729/how-to-export-your-google-calendar-events-into-microsoft-outlook/) You can sync your Google Calendar with your Microsoft Outlook in a few simple steps. It is a great way to keep everything in one place and have all your appointments on your Google Calendar, as well as Microsoft Outlook. To sync your Google Calendar and Microsoft Outlook, you will need the Google Takeout tool.
Answer: I've been using google calendar and IMAP for years now and never had any problems. I just use the standard IMAP address in Outlook and it works great.
Answer: Doesn't this already exist? http://www.outlookcalendar.com/ Although there doesn't seem to be a solution for exporting contacts, using Google Takeout (link below) allows you to export an ICS file that can be imported into Outlook. [Google Takeout](https://www.google.com/takeout) |

Table 20: The responses from two RLHF models on the StackExchange dataset.

