# OpenReview forum: "The Trickle-down Impact of Reward Inconsistency on RLHF"
_ICLR.cc/2024/Conference — ICLR 2024 poster_

### Official Review · Reviewer_n2Hx · 2023-10-30

**Soundness:** 2 fair
**Presentation:** 3 good
**Contribution:** 3 good
**Rating:** 6
**Confidence:** 4

**Summary:**

This paper investigates an important yet often overlooked problem - the robustness of the reward model (RM). The authors propose a benchmarking technique called Contrast Instructions that gauges the reward consistency of an RM. The reward consistency is measured by consistency in preference ranking if given a pair of lexically similar instructions with different ground truth responses. Concretely, it is quantified by (1) response consistency, if the RM can identify the better response for a given instruction, and (2) instruction consistency, if the RM can identify the most fitting instruction for a given response. The benchmark dataset is constructed based on four open-source human preference datasets of various NLP tasks. The authors showcased that an RLHF model trained with a more reward-consistent RM outperforms an RLHF model trained with the original RM in human evaluations.

**Strengths:**

1. This paper is well-written and it is easy to follow.
2. The idea is straightforward but the underlying research problem is significant and yet often overlooked.
3. The trickle-down effect of reward consistency on RLHF training is an interesting observation, which intuitively makes sense.
4. Table 7 is helpful in seeing that reward consistency and test set accuracy (if I understood correctly) do not necessarily correlate. This is similar in the sense that the (dis-)correlation between human score and FID is often discussed in generative models.

**Weaknesses:**

1. The way that the authors constructed the dataset, is filtering by the cosine similarity between SimCSE embeddings that are in the range of [0.75, 0.9]. This seems convenient but I wonder how reliable is this method. Have you done a manual inspection to measure the agreement rate between the method and human evaluators? Or maybe you could try using a model-based approach like prompting GPT-4?
2. As I am more interested in the benchmark dataset itself, the evaluation for dataset validation seems limited. **The author should focus on providing a reliable benchmark as the major scientific contribution**; I believe the finetuning methods discussed in the manuscript are not as significant. Therefore, the authors should provide more evaluation results on more LLMs (open-source + closed-source) to validate your benchmark dataset. The evaluation results on popular models like GPT-4 would be very helpful. Although you can't get the weights, test by prompting would be sufficient. I am also curious to know the difference in consistency between the pre-trained model vs their SFT-ed version (i.e. llama2-7b vs llama2-7b-chat). If the compute resource allows, the parameter scaling on reward consistency could also be an interesting point for investigation.
3. The benchmark dataset should be submitted along with the paper, as I believe this is the core contribution of the paper.
4. The data variances in Figure 3 make the comparison in Sec. 7.2 rather inconclusive. However, I praised the authors' honesty in showing error bars.

**Questions:**

1. How concerned are you about the risk of data leakage? What implications would arise if instances from the benchmark dataset were also present in LlaMa-7B's pre-training data or the dataset used to train the reward model? Have such overlaps impacted the assessment of reward consistency?
2. I need more details on Section 4. For "Single-Task", what is the split between train and test?
3. For Section 4, why would you fine-tune a benchmark dataset? And the fact that it performs so poorly after being trained and tested on the same dataset distribution is surprising. The consistency improvement seems marginal.
4. What is RMEval in Table 7? Test set accuracy on binary classification task?

Overall I am positive about this paper. This is an interesting piece of research. If the authors can properly address mines and other reviewer's concerns, I will agree to raise my review score.

---

> ### Author Response · Authors · 2023-11-19
>
> **Q1**:The way that the authors constructed the dataset, is filtering by the cosine similarity between SimCSE embeddings that are in the range of [0.75, 0.9]. This seems convenient but I wonder how reliable is this method. Have you done a manual inspection to measure the agreement rate between the method and human evaluators? Or maybe you could try using a model-based approach like prompting GPT-4?
>
> **A1**: Regarding cosine range, we have consideration for the following factors:
> - Upper Bound: We set the upper limit (e.g., 0.9) to avoid retrieving identical instructions. This is crucial to ensure  that $I_{A} \circ r_{A}$ is better than $I_{B} \circ r_{A}$.
>
> - Lower Bound: We avoided setting the lower bound too low (e.g., 0.5) to maintain the challenge for the RM. A lower range would make it too easy for the RM to judge, failing to test whether RM has a nuanced understanding of human preferences.
>
> - Range Interval: We didn’t opt for a narrow range (e.g., [0.85, 0.9]) as it would greatly limit the size of our Contrast Set.
>
> To satisfy all of three factors, we can only handcraft a proper range, so we set an upper limit (e.g. 0.9) to avoid retrieving identical instructions, and the lower limit (e.g. 0.75) is adjusted mostly to make sure we can sample enough test examples in each human preference dataset.
>
> We provide an automatic evaluation for the quality of our benchmarks, which uses GPT4 to evaluate our contrast instructions. Specifically, we use In-context learning (appending four demonstrations where each has a better and a worse response) to help GPT4 distinguish better or worse responses in our benchmarks. The results are shown in Appendix G. We can observe that GPT4 achieves around 95% accuracy on our benchmarks, indicating a high matching degree between our benchmark and GPT4. This guarantees the quality of our Contrast Instructions to a certain extent.
>
> Please note that our human performance (avg. 80% accuracy) on our contrast instruction does not indicate flaws in our Contrast Instructions. Since this performance is calculated under scenarios where the annotators are forbidden to use networks and search engines, the annotators’ performance would be limited to their knowledge. With the help of search engines and other tools, our annotators achieve an average performance beyond 95%.
>
> **Q2**: As I am more interested in the benchmark dataset itself, the evaluation for dataset validation seems limited. The author should focus on providing a reliable benchmark as the major scientific contribution; I believe the finetuning methods discussed in the manuscript are not as significant. Therefore, the authors should provide more evaluation results on more LLMs (open-source + closed-source) to validate your benchmark dataset. The evaluation results on popular models like GPT-4 would be very helpful. Although you can't get the weights, test by prompting would be sufficient. I am also curious to know the difference in consistency between the pre-trained model vs their SFT-ed version (i.e. llama2-7b vs llama2-7b-chat).
>
> **A2**: For GPT4’s results on our Contrast Instruction, please refer to our answer to your **Q1**.
>
> RM consistency does not equal the consistency of the final DPO-trained LLM, that’s why we use the ‘trickle-down effect’ in our title. However, exploration on SFT/RLHF LLMs would be interesting, we have explored potential methods for assessing preference consistency in LLMs trained via SFT/RLHF, each with its own limitations:
> - Prompting DPO-trained LLMs to choose the better response often leads to them not following the instruction correctly, such as repeating the instruction instead.
> - Calculating the likelihood of instruction-response pairs is hindered by length bias, with DPO-trained LLMs tending to favor longer responses, resulting in poor performance on preference benchmarks.
>
> **Q3**:The benchmark dataset should be submitted along with the paper, as I believe this is the core contribution of the paper.
>
> **A3**: We’ve just uploaded the data in the system, feel free to check.

---

> > ### Author Response · Authors · 2023-11-19
> >
> > **Q4**: How concerned are you about the risk of data leakage? What implications would arise if instances from the benchmark dataset were also present in LlaMa-7B's pre-training data or the dataset used to train the reward model? Have such overlaps impacted the assessment of reward consistency?
> >
> > **A4**: It’s a good question. In our study, specific measures to address data leakage were not the primary focus. However, we understand the importance of this issue and will ensure rigorous checks against data leakage based on the very recent work [1,2]. We agree that understanding the impact of data leakage on reward consistency is crucial, as highlighted by our use of CONTRAST INSTRUCTIONS to enhance RM consistency​​.
> >
> > But we couldn’t complete the analyses towards whether LLaMa pre-training corpus contains our data during the rebuttal period.
> >
> > [1] Detecting Pretraining Data from Large Language Models
> > [2] Proving Test Set Contamination in Black Box Language Models
> >
> > **Q5**:I need more details on Section 4. For "Single-Task", what is the split between train and test?
> >
> > **A5**: Certainly. The term 'single-task' in our study refers to training our RM exclusively on data from one particular task-specific benchmark and subsequently testing it against the Contrast Set derived from the same task. For each task, we partition the data into training, development (dev), and test sets using an 8:1:1 ratio. We then identify and utilize the checkpoint that delivers the best performance on the development set for our final evaluation.
> >
> > **Q6**:For Section 4, why would you fine-tune a benchmark dataset? And the fact that it performs so poorly after being trained and tested on the same dataset distribution is surprising. The consistency improvement seems marginal.
> >
> > A6: In Section 4, we detail the process of fine-tuning our RM using the training sets from task-specific benchmarks and then evaluating it on our Contrast Set. Our results, presented under 'single-task' and 'multi-task' training conditions, clearly indicate that multi-task training does not enhance RM consistency.
> >
> > Moreover, Table 2 shows the RM's performance on the Contrast Set, while Table 8 provides insights into its performance on the original test sets, where the RM demonstrates an average accuracy exceeding 75%. This is a huge gap that reflects how severe the RM inconsistency issue is.
> >
> > Besides, we believe that our Contrast Set and the original test sets should not be entirely viewed as the same distribution.
> > - The original benchmarks feature responses that are relatively easy to differentiate due to lower lexical similarity.
> > - Our Contrast Set comprises pairs that are lexically similar but semantically diverse, posing a significant challenge to the RM's capacity for nuanced semantic comprehension.
> >
> > This distinction is crucial as it demonstrates that achieving high accuracy on the original test sets does not necessarily equate to high performance on the more demanding Contrast Set.
> >
> > **Q7**: What is RMEval in Table 7? Test set accuracy on binary classification task?
> >
> > **A7**: RMEval refers to the performance of RM on the original testset, instead of on the Contrast set. The significant gap between these two performances shows the severe issues of RM inconsistency.
> >
> > If you have any other questions, please feel free to ask.

---

> > > ### Author Response · Authors · 2023-11-21
> > >
> > > Dear Reviewer,
> > >
> > > Please feel free to reach out if you have any additional questions or would like any clarification. We'll do our best to respond in the remaining 40 hours before the discussion period ends.

---

> ### Comment · Reviewer_n2Hx · 2023-11-22
> **Thanks for your rebuttal**
>
> Hi,
>
> I have taken a look at the newly added Appendix G and H. I praised the efforts for running these experiments during this short amount of time. The results are interesting to see - a human with a search engine performs nearly perfectly and GPT-4 performs on par with humans, while other smaller models perform nearly as random-picking. I wouldn't be able to draw any significant conclusion from the scaling tests, but I understand that it would be too much to ask you here to finetune larger models.
>
> Overall I like the idea of this work, though the rigorousness of the methods could be improved. My opinion on this paper remains the same - the authors should focus on providing a reliable benchmark supported by lots of validations instead of focusing on bringing more things into the basket of "novel contributions".
>
> Most of my questions have been resolved in this rebuttal. I agree to raise my review score. I hope the authors can continue working on this study in the future.

---

> > ### Author Response · Authors · 2023-11-22
> >
> > Thank you for taking the time to read our response and increasing the score! We will emphasize more on our benchmarking strategy in the writing of the next version.

---

### Official Review · Reviewer_Kxr1 · 2023-11-01

**Soundness:** 2 fair
**Presentation:** 2 fair
**Contribution:** 2 fair
**Rating:** 5
**Confidence:** 4

**Summary:**

The paper focuses on reward inconsistency while doing Reinforcement from Human Feedback studying the various reasons around such inconsistencies by developing relevant metrics to measure the same. They demonstrated the failure of the current reward models typically used in RLHF (with comparison to avg human)  on the CONTRAST INSTRUCTIONS introduced in the paper, which the authors attribute to reward inconsistency. Finally, the authors propose two strategies to mitigate such reward inconsistency with improved downstream performance.

**Strengths:**

The primary objective of the paper is to demonstrate the reward inconsistency with the standard RLHF training, as shown in Figure 1 where it shows for similar (but distinct) prompts, the rewards assigned are inconsistent with the current reward models. A primary reason attributed to the over-optimization is the fact that the current reward models are trained on datasets that don't represent close preferences in other words, the current reward models are not optimized for prompts where both the preferences are near-optimal and one is slightly better than the other. According to the authors, those are the places where the current reward models suffer and that's where CONTRAST INSTRUCTIONS helps in providing a meaningful evaluation and mitigation to such over-optimization.

**Weaknesses:**

1. The paper claims the issue in reward inconsistency is due to the reward over-optimization issue (citing Gao et, al). Still, the reason for such reward over-optimization and how that causes the inconsistencies in the particular Contrast Dataset is unclear.  More specifically the author claims that "From the RMs’ perspective, correctly distinguishing between a clearly good vs. bad response is easy". But the contrast dataset for example shown in Figure 1, they are very different questions although textually there are common words. Does that mean the current LLMs are not able to produce representations that can separate the two is not very clear and needs further clarification. For ex: "A is a  good student" and "A is a bad student", they have a lot of words in common but in representation space they should be extremely different and should be trivially separated if the representations are reasonable. Thus it's not very clear how Contrast Dataset is providing a challenging dataset to test RM model inconsistency.

2. Another point is that why such reward inconsistencies are not observed in standard available RLHF datasets like Carper AI, hh, etc. are not made explicit. Does it mean that the majority of the datasets are easily separable and lacks samples towards the optima which is hard for human to segregate? A comparison with current RLHF methods is critical on standard datasets to understand the significance and mitigation of the problem properly.

3. "Surprisingly, we observe close to random-chance performance ..while humans are able to rank the responses correctly in ≈ 80% of the cases" So, is it the case that humans are able to identify it but the reward models are not able to learn the same and failing is not very clear since its a supervised learning problem and can be shown to be strongly convex under certain settings. Hence, a clear description is missing on the same, and will be interesting to have a discussion with reference to the recent works showing convergence [1,2] and where this issue can arise in that context.

[1]. Banghua Zhu, Jiantao Jiao, Michael I. Jordan "Principled Reinforcement Learning with Human Feedback from Pairwise or K-wise Comparisons"

[2]. Souradip Chakraborty, Amrit Singh Bedi, Alec Koppel, Dinesh Manocha, Huazheng Wang, Mengdi Wang, Furong Huang "PARL: A Unified Framework for Policy Alignment in Reinforcement Learning"

**Questions:**

1. Why augmentation helps as a solution to mitigate the problem is not clear in the context of the Contrast Dataset. Will be helpful to have a more rigorous discussion on the same?
2. ContrastDataset provides is challenging dataset, how are humans able to do good on it? Is it mainly hard for the LLMs since the representations are sub-optimal?
3. A comparison with current SOTA RLHF or reward models on standard/open-sourced datasets will be helpful in understanding the crux of the problem.
4. Recent work on Direct Pref Optimization (DPO) shows that it can learn with learning reward models, will such an issue happen there as well?
5. Reward ensemble seems to work well in RLHF to mitigate over-optimization as also followed in [1] for Robotics. Will be interesting to see if that helps and have a discussion around the same.
6. The notations used in the equation after 1 (missing no) are not very clear and it will be helpful if they can be updated as per standard RL notations of trajectory, state, etc.

[1]. Thomas Coste, Usman Anwar, Robert Kirk, David Krueger "Reward Model Ensembles Help Mitigate Overoptimization " https://arxiv.org/abs/2310.02743

---

> ### Author Response · Authors · 2023-11-19
>
> Thanks you for your comments and questions. We hope our responses can resolve your concerns:
>
> **Q1**: The paper claims the issue in reward inconsistency is due to the reward over-optimization issue (citing Gao et, al). Still, the reason for such reward over-optimization and how that causes the inconsistencies in the particular Contrast Dataset is unclear. More specifically the author claims that "From the RMs’ perspective, correctly distinguishing between a clearly good vs. bad response is easy". But the contrast dataset for example shown in Figure 1, they are very different questions although textually there are common words. Does that mean the current LLMs are not able to produce representations that can separate the two is not very clear and needs further clarification. For ex: "A is a good student" and "A is a bad student", they have a lot of words in common but in representation space they should be extremely different and should be trivially separated if the representations are reasonable. Thus it's not very clear how Contrast Dataset is providing a challenging dataset to test RM model inconsistency.
>
> **A1**:
>
> Please note that we are **NOT** claiming the specific benchmark itself is challenging for all reward models or LLMs. What we are claiming here is the **Contrast Instructions** benchmarking strategy, when applied to a specific RM training set to create an evaluation set, can reveal inconsistencies of the resulting RMs.
>
> First, we want to clarify that the (short) example shown in Figure 1 is mostly for illustration purposes due to space constraints. Most examples in contrast instructions involve much longer responses that are difficult for the models to distinguish – some of the examples can be found in Table 1.
>
> But even with the “A is a good student” vs. “A is a bad student” example, the representation would be “extremely different” **only when your model is properly trained to generalize to the task that it’s solving**. e.g. sentiment classification. Let’s say if the task prompt asks “how topically relevant are the two sentences”, the two would have similar representations. Here you are making the assumption that the model generalizes well from training, and so it would be sensitive to semantic changes even when there’s a lot of lexical overlap between responses. What we intend to show with our $C_{res}$ and $C_{ins}$ metric is exactly that RMs do NOT generalize well with the standard training objective. In NLP, similar generalization issues and insensitivity to examples with high lexical overlap has been observed across different settings [e.g. 1, 2]. We are observing similar patterns with RMs as well.
>
> Reward Over-optimization – For example, within the open-source community (e.g. w/ Llama1 and 2), examples from StackExchange make up for majority of the RM training data.  StackExchange follows a specific “proxy” way of defining human preferences, e.g. within the same question, each human preferences example are constructed from responses to same question/thread, and responses with the highest votes are considered the preferred responses. What we intend to show with our $C_{res}$ metric is that RMs trained this way only fits the training distribution, and don’t generalize to what we think RM should be doing – i.e. being able to distinguish between good/bad responses consistently. With contrast instructions, we observe that RMs fail to do so even when the questions come from the domain (e.g. StackExchange) as seen in training.
>
> [1] Gardner, Matt, et al. "Evaluating Models’ Local Decision Boundaries via Contrast Sets." Findings of the Association for Computational Linguistics: EMNLP 2020. 2020.
>
> [2] McCoy, Tom, Ellie Pavlick, and Tal Linzen. "Right for the Wrong Reasons: Diagnosing Syntactic Heuristics in Natural Language Inference." Proceedings of the 57th Annual Meeting of the Association for Computational Linguistics. 2019.

---

> > ### Author Response · Authors · 2023-11-19
> >
> > **Q2**: Another point is that why such reward inconsistencies are not observed in standard available RLHF datasets like Carper AI, hh, etc. are not made explicit. Does it mean that the majority of the datasets are easily separable and lacks samples towards the optima which is hard for human to segregate? A comparison with current RLHF methods is critical on standard datasets to understand the significance and mitigation of the problem properly.
> >
> > **A2**:
> > Thanks for your question. We're claiming the specific benchmark itself is challenging for reward models. What we are claiming here is the **Contrast Instructions** benchmarking strategy.
> >
> > Moreover, we don’t claim open benchmarks are flawed, we don’t observe consistency issues on them due to the following differences.
> > In our contrast set, we use golden response from a similar instruction as a negative example, which can be regarded as out-of-distribution samples and make RM difficult to distinguish.
> > In standard benchmarking, their testset are more in-domain datasets compared to training dataset, which makes RM easier to recognize.
> >
> > Regarding open benchmarks and models, as specified in our experimental settings, we use the official RM checkpoints (Stack-LLaMa) from Huggingface RLHF team (https://huggingface.co/trl-lib/llama-7b-se-rm-peft), which is trained on open-source benchmark Stack-Exchange. Our results show that it still suffers from severe inconsistency issues.
> >
> > **Q3**: Surprisingly, we observe close to random-chance performance ..while humans are able to rank the responses correctly in ≈ 80% of the cases" So, is it the case that humans are able to identify it but the reward models are not able to learn the same and failing is not very clear since its a supervised learning problem and can be shown to be strongly convex under certain settings. Hence, a clear description is missing on the same, and will be interesting to have a discussion with reference to the recent works showing convergence [3,4] and where this issue can arise in that context.
> >
> > [3]. Banghua Zhu, Jiantao Jiao, Michael I. Jordan "Principled Reinforcement Learning with Human Feedback from Pairwise or K-wise Comparisons"
> >
> > [4]. Souradip Chakraborty, Amrit Singh Bedi, Alec Koppel, Dinesh Manocha, Huazheng Wang, Mengdi Wang, Furong Huang "PARL: A Unified Framework for Policy Alignment in Reinforcement Learning"
> >
> > **A3**: From a generalization point of view, what our metric $C_{res}$ is measuring is not what the model is being trained on exactly. See our response above for “Reward Over-optimization”. It’s a supervised learning problem, but again, what we are claiming here is the poor generalization abilities with RMs trained under this protocol.
> >
> > As for the derivations from the theoretical works in [3,4] or convexity in your comment, we believe they are quite distant from real-world scenarios:
> > - Non-convexity. In the context of RM, we aim to optimize towards human preference. Optimizing towards human preference is more than being a convex optimization problem, which has shown in tons of works in RLHF literature.
> > - Misaligned targets: [4] aims to propose a framework that tackles challenges in policy alignment in reinforcement learning, instead of reward modeling, which is the main focus in our paper. Despite its contributions, theoretical analysis in [4] is not very helpful to give insights about the reward model we discuss in this paper.
> > - Misaligned targets: [3] shows that when the true reward function is linear (in their abstract and intro), the widely used maximum likelihood estimator (MLE) converges under both the Bradley-Terry-Luce (BTL) model and the Plackett-Luce (PL) model. However, we know that the reward model trained from human preference is non-linear.
> >
> > - Strong assumptions. [3] assumed that the policy trained is greedy with respect to the learned reward. However, in practice the reward is mostly used to fine-tune the pre-trained policy, as claimed in their conclusions.
> >
> > **Q4**: Why augmentation helps as a solution to mitigate the problem is not clear in the context of the Contrast Dataset. Will be helpful to have a more rigorous discussion on the same?
> >
> > **A4**: In Section 7.2, we delve into how our methods address the inconsistency problem by analyzing it from a model calibration perspective. This approach offers a comprehensive understanding of the underlying issues and potential solutions.

---

> ### Author Response · Authors · 2023-11-19
>
> **Q5**: ContrastDataset provides is challenging dataset, how are humans able to do good on it? Is it mainly hard for the LLMs since the representations are sub-optimal?
>
> **A5**: While Reward Models (RMs) currently struggle to differentiate patterns within Contrast Instructions, this challenge is not mirrored in human performance. Humans are adept at discerning subtle semantic differences, an ability that RMs have yet to develop fully.
>
> This disparity highlights the difficulty RMs face in learning representations that effectively separate human-preferred responses from non-preferred ones. At least, our results show the limitation of the main-stream protocol for modeling RM, thus motivating more sophisticated designs for RM training.
>
> **Q6**: Recent work on Direct Pref Optimization (DPO) shows that it can learn with learning reward models, will such an issue happen there as well?
>
> **A6**: Given that DPO algorithms do not incorporate an explicit RM, evaluating their consistency in the same manner as RMs is not feasible.
>
>
> Moreover, RM consistency does not equal the consistency of the final DPO-trained LLM. That’s why we use ‘trickle-down effect’ in our title. However, we have explored potential methods for assessing preference consistency in LLMs trained via DPO, each with its limitations:
> - Prompting DPO-trained LLMs to choose the better response often leads to them not following the instruction correctly, such as repeating the instruction instead.
> - Calculating the likelihood of instruction-response pairs is hindered by length bias, with DPO-trained LLMs tending to favor longer responses, resulting in unfair evaluation on preference benchmarks.
>
> **Q7**: Reward ensemble seems to work well in RLHF to mitigate over-optimization as also followed in [1] for Robotics. Will be interesting to see if that helps and have a discussion around the same.
>
> **A7**: We tested the efficacy of ensembling various Reward Models (RMs) using checkpoints from Stack-exchange available on Huggingface, including
> - https://huggingface.co/trl-lib/llama-7b-se-rm-peft
> - https://huggingface.co/mnoukhov/llama-7b-se-rm-peft
> - Our trained LLaMa-7B reward model (On Stack-exchange)
> - Our trained LLaMa2-7B reward model (On Stack-exchange)
>
>
> Our experiments incorporated three ensembling techniques referenced in the paper [5]: mean, worst, and uncertainty-based ensembling. However, as shown in our results below, this approach did not yield significant improvements over our baseline model. Overall, there is no statistical significance between performance of three ensembling methods, based on a pair-wise t-test where p value is not smaller than 0.05. This outcome indicates that simply ensembling different RMs can not address the core issues we are investigating.
>
> |  | None | Mean | Worst-Case | Uncertainty-Weighted |
> |---|---|---|---|---|
> | Performance | 50.1 | 49.2 | 50.3 | 50.6 |
>
> [5]: Reward Model Ensembles Help Mitigate Overoptimization
>
> If you have any other questions, please feel free to ask.

---

> > ### Author Response · Authors · 2023-11-21
> >
> > Dear Reviewer,
> >
> > Please feel free to reach out if you have any additional questions or would like any clarification. We'll do our best to respond in the remaining 40 hours before the discussion period ends.

---

> > ### Comment · Reviewer_Kxr1 · 2023-11-22
> > **Response to the Comment by Authors**
> >
> > Thanks for the rebuttal and providing details to my questions. I have a few more clarification questions to better understand the proposed method
> >
> > 1. Regarding Q2, I am still not very sure that the representation learned by the LLMs are unable to distinct  “A is a good student” vs. “A is a bad student”, and also the references provided are of 2020 and before, so not sure if it applies to the evaluation performance of the SOTA LLMs, will be helpful to provide some latest evaluation studies if any (Poor generalization of SOTA LLMs )?. I agree that very challenging or complex contrast examples, it can be a concern but are there some examples of such, will be helpful to point there (sorry in case I might have overlooked them).
> >
> > 2. Also, I didn't quite understand the answer to Q3, that if humans have given the correct preferences, still the reward models can't learn is because of the poor generalization of reward models? What is the specific reason? I see that reward overoptimization is used in the context? Can you please explain specifically what the relationship between the two is with concrete references?
> >
> > 3. Also, the reason for the references was to understand which component of the analysis might break based on your hypothesis, which is not clear. "The reward model trained from human preference is non-linear." ---> Its important to note that it has been considered linear [3] in the representation space. We know that pre-trained SOTA LLM embeddings provide a good representation space, and linearity on that might be restrictive I agree but not an unreasonable assumption to study. To summarize, a more rigorous discussion is required which will help in understanding both points 1 and 2.
> >
> > 4. "DPO-trained LLMs tending to favor longer responses," I guess that is also a problem with the current RLHFs as shown in [1]. Hence, it's not clear why this concern is appearing only for DPO and not for RLHF.
> >
> > [1]. Prasann Singhal, Tanya Goyal, Jiacheng Xu, Greg Durrett . A Long Way to Go: Investigating Length Correlations in RLHF https://arxiv.org/abs/2310.03716

---

> > > ### Author Response · Authors · 2023-11-22
> > >
> > > Thank you for taking the time to read our response and engage in discussion! Based on your further questions, here are our responses.
> > >
> > > Q1:
> > > >... I am still not very sure that the representation learned by the LLMs are unable to distinct “A is a good student” vs. “A is a bad student”...
> > >
> > > A1: First, we want to clarify that learning a reward model is not for classification but regression. The standard protocol tries to tackle this regression problem by classification loss. An optimal RM should assign reward scores to different instruction-response pairs that align perfectly (e.g., Spearman correlation=1.0) with their quality (judged by humans). Thus, previous experiences of embedding-based classification could not be well transferred to the RM case.
> > >
> > > Concretely, RM is trained to rank different instruction-response pairs $I_{1}+r_{1}$, $I_{1}+r_{2}$,..., $I_{1}+r_{N}$ (assume that $r_{1}$ is the golden response). In our case, our benchmarking strategy creates $I_{1}+r_{*}$ (never seen in the training set) guaranteed with worse quality than $r_{1}$, which causes a failure for RM. The issue shown in our paper is not caused by LLM itself (which is also not our claim) but by the standard and commonly accepted protocol of RM training. This can be due to this protocol's model architecture, training objective, and other elements. The BERT-era models also encounter more challenges when facing ranking problems than classification cases. For example, XLNet achieves only 20.28 ERR@20 on ClueWeb09-B (a document ranking task) [a]. Furthermore, we believe the LLaMa-7B used in our paper is stronger than the BERT-era models.
> > >
> > >
> > > Besides, our phenomenon has been observed in more recent work [b], where RM performs randomly on their 'adversarial sets.'
> > >
> > > [a] Zhilin Yang, Zihang Dai, Yiming Yang, Jaime Carbonell, Ruslan Salakhutdinov, Quoc V. Le
> > > "XLNet: Generalized Autoregressive Pretraining for Language Understanding."
> > > https://proceedings.neurips.cc/paper/2019/file/dc6a7e655d7e5840e66733e9ee67cc69-Paper.pdf
> > >
> > > [b] Zhiyuan Zeng, Jiatong Yu, Tianyu Gao, Yu Meng, Tanya Goyal, Danqi Chen  "Evaluating Large Language Models at Evaluating Instruction Following" https://arxiv.org/abs/2310.07641
> > >
> > >
> > > **Q2**:
> > > > …if humans have given the correct preferences, still the reward models can't learn is because of the poor generalization of reward models….
> > >
> > > **A2**:
> > > We provide an explanation for such a failure from the perspective of model calibration in Section 7.2, where the RMs fail to do well in calibration.
> > >
> > > Overall, we believe that the RM issues in our paper are not caused by poor generalization of LLM (which is also not our claim) but by the standard and commonly accepted protocol of RM training. This can be due to the model architecture, training objective (we have applied margin-based loss from LLaMa2 to train RM but failed to solve the inconsistency issues, as shown in our response to Reviewer iZzr), and other elements. The exact reason for this remains to be explored in future work. Our paper focuses on providing an automatic benchmarking strategy to expose these issues and some simple methods to resolve these issues to a certain extent.
> > >
> > > **Q3**:
> > > >...Also, the reason for the references was to understand which component of the analysis might break based on your hypothesis, which is not clear... Its important to note that it has been considered linear [3] in the representation space.
> > >
> > > **A3**: The discussion can refer to our answers to Q1 and Q2, where we show the key differences between embedding-based classification and RM training.
> > >
> > > Besides, we want to point out the claims in [3]. [3] is not claiming that the reward model is linear or non-linear but provides generalization bounds if the RM is linear (In the main paper) or non-linear (In Appendix A) under certain assumptions. Despite its significant theoretical contributions, the setup in [3] has a gap in realistic LLM-based RM training. Thus, some assumptions/conclusions in [3] can not be transferred to real-world scenarios.
> > >
> > > [3]. Banghua Zhu, Jiantao Jiao, Michael I. Jordan "Principled Reinforcement Learning with Human Feedback from Pairwise or K-wise Comparisons"  https://arxiv.org/abs/2301.11270
> > >
> > > **Q4**:
> > > >..."DPO-trained LLMs tending to favor longer responses," I guess that is also a problem with the current RLHFs as shown in [1]. Hence, it's not clear why this concern is appearing only for DPO and not for RLHF...
> > >
> > > **A4**: We didn’t claim that RLHF does not suffer from this issue; we know that RLHF also suffers from this issue (In fact, our response to Reviewer n2Hx also mentioned that RLHF would suffer from this). In that response, we answer why explicitly examining RM in DPO would be empirically implausible, so we only focus on the properties of DPO-trained models.

---

> > > > ### Comment · Reviewer_Kxr1 · 2023-11-22
> > > > **Response to Comment by Authors**
> > > >
> > > > Thanks for the detailed response to my comments. It helps to clarify some of my confusion and hopefully will improve the draft for better readability.
> > > >
> > > > > "Overall, we believe that the RM issues in our paper are not caused by poor generalization of LLM (which is also not our claim) but by the standard and commonly accepted protocol of RM training. This can be due to the model architecture, training objective (we have applied margin-based loss from LLaMa2 to train RM but failed to solve the inconsistency issues, as shown in our response to Reviewer iZzr), and other elements. The exact reason for this remains to be explored in future work. "
> > > >
> > > > So, if I understand correctly as the author states, the generalization of LLMs is not the reason for such issues arising in the paper which is reasonable, and I also believe that should not be the issue. On the other hand, it's still unclear what the reason actually is -->"model architecture, training objective, and other elements". Since it is critical to understand the reason for being able to reliably comment on the contribution of the paper. Hence, an ablation will be important to understand the reason behind this and why in spite of humans providing correct labels (so y is correct), and LLMs providing correct representation space (x, x-->y mapping is correct) a supervised model is not able to learn and generalize. Thus ablation in model architectures and on more reliable benchmarks will be critical.

---

> > > > > ### Author Response · Authors · 2023-11-23
> > > > >
> > > > > Hi, thanks for your suggestions; we selected three different model architectures
> > > > > - Decoder-only: LLaMa-7B in our paper.
> > > > > - Encoder-only: Roberta-Large
> > > > > - Encoder-decoder: google/flan-t5-base
> > > > > Here are their performance on our benchmarks.
> > > > >
> > > > > |  | Stack | WMT | Twitter | Sum |
> > > > > |:---:|:---:|:---:|:---:|:---:|
> > > > > | LLaMa | 50.1 | 56.4 | 60.1 | 47.7 |
> > > > > | RoBerta | 50.8 | 55.8 | 57.6 | 48.3 |
> > > > > | Flan-t5| 51.2 | 57.3 | 60.3 | 48.9 |
> > > > >
> > > > > As shown in the results, **simply** changing the model architecture is not significantly helpful in resolving the consistency issues. We will add these ablations to the next version. Still, we could not test more models due to the closing rebuttal deadline, and we share some extra thoughts based on your arguments.
> > > > >
> > > > >
> > > > >
> > > > > ## The relation between **rebuilding standard protocol of RM training** and **our paper**
> > > > > Although proposing new protocols for RM training is undoubtedly important and is promising future work, it is not **the main focus of our paper, and we make no strong claims against it**.
> > > > > Also, we want to reclaim the main contributions in our paper, as shown in Section 1 of our paper.
> > > > > - We define what should be regarded as consistency/inconsistency for RMs and propose an automatic benchmarking strategy (Contrast Instruction) tailored for consistency evaluation.
> > > > > - We discover that the RMs trained by mainstream methods perform catastrophically (akin to random guessing) on our Contrast Instruction, which is task-agnostic and model-size-agnostic.
> > > > > - We try to design some simple methods (without pre-assuming the patterns of Contrast Instruction for fair evaluation) to enhance consistency but bring very limited improvements, which emphasizes the challenge of these issues.
> > > > > - Finally, we measure the trickle-down effects of reward consistency. A more consistent RM indeed benefits the final RLHF-trained model.
> > > > >
> > > > >
> > > > > > If someone wants to rebuild the protocol for RM training, our paper can be regarded as a cornerstone for
> > > > > - showing strong evidence for the drawbacks of the current RM training protocol that motives their work.
> > > > > - providing an automatic benchmarking strategy for more challenging tests of RM evaluation.
> > > > >
> > > > > ## Discussion for RM quality
> > > > > Moreover, we also want to clarify that whether an RM is good is also (not only reflected in standard benchmarks) determined by the final results of RLHF (apply such an RM for RLHF). That is why we name our title "The **trickle-down effects** of reward consistency in RLHF" since we should not discuss the quality of RM being isolated from the ultimate results of RLHF, as detailed in our paper.

---

> > > > > > ### Comment · Reviewer_Kxr1 · 2023-11-23
> > > > > > **Response to Comment by Authors**
> > > > > >
> > > > > > Thanks for the response. I agree with most of the points mentioned in the final response and definitely would request the authors to add the points of discussion to the paper. I will increase my score, but still, as I mentioned more technical clarity is needed with further ablation with more emphasis on the point of why the reward model is unable to learn even when human preferences are accurate and also LLM representations are generalizable. Some of these points are mentioned during our discussion which if added will help the readers to understand the significance.

---

### Official Review · Reviewer_iZzr · 2023-11-05

**Soundness:** 2 fair
**Presentation:** 2 fair
**Contribution:** 3 good
**Rating:** 5
**Confidence:** 5

**Summary:**

The paper studies reward models in RLHF under the lens of consistency -- whether the RM can adapt its scores to semantic changes to different prompt response pairs. The authors create benchmarks to verify consistency of rewrd models called Contrast Instruction where the authors study the reward model score for lexically similar instructions with different responses. They claim that current RM with the standard loss functions suffer in these contrast instructions compared to human preferences. The authors provide two techniques -- one at train time and one at inference time that improve reward consistency through extrapolation across similar examples. They also claim that reward model consistency has a correlation with the usefulness of the RLHF responses.

**Strengths:**

The paper exposes an interesting and useful aspect of reward modeling in RLHF -- reward model consistency. Apart from standard RM eval, evaluating RMs for consistency is an important aspect that future works can take into consideration or the current eval sets be expanded to include this as a benchmark task. The authors clearly define what consistency means and setup constrast instructions dataset to evaluate the reward models. While the dataset needs to be vetted more carefully, having this dataset as part of standard RM evals could be a useful exercise for practitioners. The methods that the authors propose to improve consistency are simple to implement.

**Weaknesses:**

While the paper exposes an important aspect of reward models, one thing that could be improved about in  the paper is the limited nature of experiments and evaluation.

*  I was not sure of whether the contrast instructions generated automatically do mean something different where one answer is strictly better than the other -- for instance . The authors say they restricted this based on a particular cosine distance range, but some more rigorous evaluation around this could have made the impact of the dataset a bit stronger. For instance, one can query a bigger open LLMs in order to validate that the contrast instructions are fit for purpose. The other option could have been to conduct human eval on a randomly selected subset of the contrast instructions dataset.

*  The authors could have just chosen one single baseline LLAMA 7B and use that to validate their claims. The behaviour of LLAMA 7B could be ground in the kinds of datasets and training procedures that it was subject to. Having few other RM baselines trained with similar RM training loss could have made the claims even more stronger.

 *  The human evaluation was done on 100 randomly selected data points and no standard errors and variance numbers were reported around the results. Since some of the results are close to each other, having the error bands around the result should help clarify the significance of improvements.

One other concern about human evaluation that I have is about the possibilities of potential bias as the authors themselves serve as human annotators for human eval of the results. There are several important statements made in this paper based on human evaluation. I would have preferred at least a mixture of external annotators (who are not aware of the work or how the responses were generated) to de-bias the evaluations.
"Finally, we report human performance resulting from the majority vote of three human annotators (the authors) on 100 randomly selected data points."

**Questions:**

I have the following questions for the authors

1. There were certain choices of methods made in terms of methods for response/instruction similarity and the dataset construction including the augmented data points for ConvexDA. Have other methods been evaluated and ablated against before choosing these methods ? I wanted to understand if we are doing the best we can in terms of construction of these datasets and methods.

2. I know you have considered the classic RM loss, log (sigmoid (R_chosen - R_rejected)) ? Have you considered other loss functions such as the margin loss used in LLAMA 2 ? I am not specifically looking for one kind of loss, but just a choice of few other RM losses. Considering different losses can help us understand whether RM inconsistency arises from the dataset or the kind of losses used or a combination of both. This would be a good ablation to understand where RM developers should focus more of their efforts on.

3. Lot of times, RM training can easily lead to overfitting to the training datasets and creating a generalization gap in the process. Can you kindly explain how you do model selection for RM training for the baseline and your variants ?

4. I know that you choose only a single example in ConvexDA while you construct a few semantically similar examples. Have you evaluated what happens if you include all of them ? I know this is unfair to be compared to the baseline, but it would help us understand having how many semantically similar examples will be useful.

5.In table 2, I was not sure what (estimated) human performance means. Can you kindly clarify ?

6. I know evaluating with larger models would be resource intensive. But would it be possible to run the analysis with either a smaller or larger model than 7B to understand if and how reward model sizes influences reward consistency.

---

> ### Author Response · Authors · 2023-11-19
>
> Thanks for your efforts in reviewing our paper. Here are the responses that we hope to resolve your concerns.
>
> **Q1**: I was not sure of whether the contrast instructions generated automatically do mean something different where one answer is strictly better than the other -- for instance . The authors say they restricted this based on a particular cosine distance range, but some more rigorous evaluation around this could have made the impact of the dataset a bit stronger. For instance, one can query a bigger open LLMs in order to validate that the contrast instructions are fit for purpose. The other option could have been to conduct human eval on a randomly selected subset of the contrast instructions dataset.
>
> **A1**: Thank you for the comments. We have human evaluation results on $C_{res}$ on each dataset in table 8 of Appendix B. Due to space constraints, we only reported human performance averaged over 4 datasets in Table 2.
>
> After reading the reviews, we conducted a more thorough human evaluation on each of the dataset. On average over 4 datasets , we see that humans rank responses correctly ~82% of the time. If we allow for Google search to access relevant knowledge, the accuracy goes up to 96%. We also conducted the same evaluation with GPT4 prompting with 4-shot demonstrations, from which we observed 95% accuracy in terms of $C_{res}$. We think the gap between RM vs. human performance helps justify our observations on RM inconsistency.
>
> Regarding cosine range, we have consideration for the following factors:
> - Upper Bound: We set the upper limit (e.g., 0.9) to avoid retrieving identical instructions. This is crucial to ensure  that $I_{A} \circ r_{A}$ is better than $I_{B} \circ r_{A}$.
>
> - Lower Bound: We avoided setting the lower bound too low (e.g., 0.5) to maintain the challenge for the RM. A lower range would make it too easy for the RM to judge, failing to test whether RM has a nuanced understanding of human preferences.
>
> - Range Interval: We didn’t opt for a narrow range (e.g., [0.85, 0.9]) as it would greatly limit the size of our Contrast Set.
>
> To satisfy all of three factors, we can only handcraft a proper range, so we set an upper limit (e.g. 0.9) to avoid retrieving identical instructions, and the lower limit (e.g. 0.75) is adjusted mostly to make sure we can sample enough test examples in each human preference dataset.
>
> We have updated the human eval and GPT4 results in Appendix G, where both human and GPT4 obtain over 90% accuracy on our Contrast Instruction. We believe these results can serve as a quality check for the data built by our Contrast Instruction benchmarking strategy. Moreover, we’ll try to allot more space to show human performance in the main body as well.
>
>
> **Q2**: The authors could have just chosen one single baseline LLAMA 7B and use that to validate their claims. The behaviour of LLAMA 7B could be ground in the kinds of datasets and training procedures that it was subject to. Having few other RM baselines trained with similar RM training loss could have made the claims even more stronger.
>
> **A2**: Thanks for you this suggestion! – In Appendix H, we've added experiments with RMs trained with backbone models of different sizes. We observe that scaling model size leads to (1) increased test performance on the original human preference dataset, but (2) the $C_{res}$ and $C_{ins}$ performance remains flat. This echoes with our hypothesis that – standard RM training fits the only the training distribution, but do not lead to a generalized model that can distinguish good/bad responses consistently. Just like what our general response said, this RM consistency issue is model-agnostic.
>
> **Q3**: The human evaluation was done on 100 randomly selected data points and no standard errors and variance numbers were reported around the results. Since some of the results are close to each other, having the error bands around the result should help clarify the significance of improvements. One other concern about human evaluation that I have is about the possibilities of potential bias as the authors themselves serve as human annotators for human eval of the results. There are several important statements made in this paper based on human evaluation. I would have preferred at least a mixture of external annotators (who are not aware of the work or how the responses were generated) to de-bias the evaluations.
>
> **A3**: Thank you for your comments.
>
> Regarding your suggestion, we recruit two emergency Computer Science expert annotators (as understanding StackExchange responses requires domain knowledge in CS). The new results are shown in Appendix J, which aligns with our human evaluation. Besides, we provide the Cohen kappa correlation to show consistency between human annotation, which show that these two human evaluations share a certain agreement.

---

> > ### Author Response · Authors · 2023-11-19
> >
> > **Q4**: There were certain choices of methods made in terms of methods for response/instruction similarity and the dataset construction including the augmented data points for ConvexDA. Have other methods been evaluated and ablated against before choosing these methods ? I wanted to understand if we are doing the best we can in terms of construction of these datasets and methods.
> >
> > **A4**: In our definition of inconsistency, only when reward scores shift to the opposite direction of quality changes can be considered inconsistent issues.
> > When constructing a benchmark for evaluating RM consistency, there are mainly two challenges:
> > - (1) guarantee: we need to guarantee that $I_{A} \circ r_{A}$ has higher qualities than $I_{A} \circ r_{B}$.
> > - (2) automation: we need to find an affordable and scalable benchmarking strategy.
> >
> > We have tried other alternatives.
> >
> > (a) Doing data augmentation on responses/instructions (i.e., as suggested by you), but it is difficult to guarantee how the quality changes after data augmentation, which breaks (1).
> >
> > (b) Recruit human experts to annotate the Contrast Instructions. This would be ideal, but it remains too expensive considering our limited budget, which breaks (2).
> >
> > **Q5**: I know you have considered the classic RM loss, log (sigmoid (R_chosen - R_rejected)) ? Have you considered other loss functions such as the margin loss used in LLAMA 2 ? I am not specifically looking for one kind of loss, but just a choice of few other RM losses. Considering different losses can help us understand whether RM inconsistency arises from the dataset or the kind of losses used or a combination of both. This would be a good ablation to understand where RM developers should focus more of their efforts on.
> >
> > **A5**: Thanks for your suggestion! We retrain RM using margin loss (adding margin $r_{m}$ just like LLaMa2 paper), where the margin is selected based on the dev set. The results are shown as follows:
> >
> > |  | Stack | WMT | Twitter | Sum |
> > |:---:|:---:|:---:|:---:|:---:|
> > | no margin | 50.1 | 56.4 | 60.1 | 47.7 |
> > | with margin | 51.1 | 56.8 | 59.8 | 50.1 |
> >
> > The results show that the margin loss provides very limited improvements to the baseline. Overall, we believe the challenge of reward consistency is relevant to various factors, including model architecture, loss function, data, etc.
> >
> > **Q6**: Lot of times, RM training can easily lead to overfitting to the training datasets and creating a generalization gap in the process. Can you kindly explain how you do model selection for RM training for the baseline and your variants?
> >
> > **A6**: In our experiments, we prepare a devset for each benchmark. We select the checkpoint that achieves the optimal result on the dev set. Specifically, the dev set is randomly sampled and split from the original test set. Moreover, we also have a devset for our Contrast set, based on our observations, the checkpoints generally perform badly on the devsets of the Contrast Set.
> >
> > **Q7**: I know that you choose only a single example in ConvexDA while you construct a few semantically similar examples. Have you evaluated what happens if you include all of them ? I know this is unfair to be compared to the baseline, but it would help us understand having how many semantically similar examples will be useful.
> >
> > **A7**: We have a detailed ablation study in Appendix F.1 for ConvexDA. If we’re going to use all the augmented data, then ConvexDA degenerates to the case of standard data augmentation (DA). In Appendix F.1, we present a comparison between ConvexDA and standard data augmentation (DA) techniques, such as paraphrasing. ConvexDA is not merely aimed at outperforming traditional data augmentation methods in terms of raw performance. Instead, its core advantage lies in offering a better trade-off between performance and efficiency. Figure 8 shows that ConvexDA achieves the desired augmentation effect with better efficiency.
> >
> > **Q8**: In table 2, I was not sure what (estimated) human performance means. Can you kindly clarify ?
> >
> > **A8**: The estimated human performance was derived from a majority vote among the annotators on a subset of data points, due to our limited budget. Therefore, considering being rigorous, we say estimated human performance here.

---

> > > ### Author Response · Authors · 2023-11-21
> > >
> > > Dear Reviewer,
> > >
> > > Please feel free to reach out if you have any additional questions or would like any clarification. We'll do our best to respond in the remaining 40 hours before the discussion period ends.

---

### Official Review · Reviewer_gxZU · 2023-11-09

**Soundness:** 3 good
**Presentation:** 3 good
**Contribution:** 3 good
**Rating:** 6
**Confidence:** 3

**Summary:**

This paper studies the problem of Reward Model (RM) inconsistency and its impact on the downstream RLHF model. The paper introduces the "Contrast Instruction" benchmark to measure the ability of a reward model to consistently recognize semantic changes in prompts and adapt reward assignments accordingly. It then shows that current RMs trained with standard ranking objectives underperform compared to human judgment on this benchmark. To address this inconsistency issue, the paper proposes two innovative techniques: "ConvexDA" and "RewardFusion," which leverage extrapolation during RM training and inference, respectively, to enhance RM consistency without requiring additional training resources. The authors demonstrate that these advancements lead to more useful responses from RLHF models trained with a more consistent RM, highlighting the crucial role of consistency in maximizing the effectiveness of the RLHF process.

**Strengths:**

1. The paper presents a novel way to investigate inconsistency in reward model via constructing benchmarks with inconsistent instruction and response pairs. The paper further proposes a fine-tuned reward model on top of the contrastive instruction-response pairs, which leads to improved RLHF performance. I think the investigation sheds insight in the RLHF research field and the proposed method is neat and effective.

2. The authors have performed extensive empirical study of the inconsistency of RMs, which clearly show that the current RMs do suffer from inconsistencies. The paper also provides thorough empirical evidence that shows the RLHF model can improve a lot with a more consistent RM trained on the constructed contrastive pairs of instruction and response.

**Weaknesses:**

1. It's a little unclear why CONVEXDA is used as a way to robustify the RM. It seems that it might work because it specifically targets the proposed benchmark, which mainly tests inconsistent instruction-response pairs with lexically similar words, but I'm not sure if the method can work well in scenarios where inconsistent responses/instructions are beyond just lexically similar words with different meanings. Moreover, I wonder if it would work equally well to simply use a different LLM to generate those similar pairs for augmentation.

2. I'm also less clear about why REWARDFUSION is needed and how it contributes to fixing inconsistency. Again, it seems to mainly target the type of inconsistency due to lexically similar words with different meanings, which may not be general enough.

**Questions:**

1. Please clarify how general the CONVEXDA and REWARDFUSION are in fixing inconsistency beyond lexically similar but different words.
2. Please compare to methods that simply use LLMs for data augmentation.

---

> ### Author Response · Authors · 2023-11-19
>
> Thank you for your comments and questions! Here are the response that we hope to resolve your concerns.
>
> **Q1**: It's a little unclear why CONVEXDA is used as a way to robustify the RM. It seems that it might work because it specifically targets the proposed benchmark, which mainly tests inconsistent instruction-response pairs with lexically similar words, but I'm not sure if the method can work well in scenarios where inconsistent responses/instructions are beyond just lexically similar words with different meanings.
>
> **Q2**:I'm also less clear about why REWARDFUSION is needed and how it contributes to fixing inconsistency. Again, it seems to mainly target the type of inconsistency due to lexically similar words with different meanings, which may not be general enough.
>
> **A1+A2**: One can embed lexical variations in training when training RMs, but that is not the point. Our goal was to introduce methodologies to improve RM consistency in a manner that is **agnostic** to the choice of evaluation (i.e., not presupposing any knowledge of the patterns inherent in Contrast Instructions), thereby ensuring a fair evaluation.
>
> Our proposal approaches (ConvexDA and RewardFusion) are not informed of the pattern Contrast Instructions, and serve as general augmentation methods. Moreover, they both contribute to performance enhancements on the original test sets, as evidenced in Table 7 (RMEval) and Table 8 (Original test set), indicating that they are beneficial beyond the Contrast set pattern. Therefore, our approaches are designed not to be informed by the patterns specific to Contrast Instructions. They are crafted to address the general generalization of RM, like other data augmentation tricks. This overarching goal explains why the improvements (on the Contrast set) attributed to these methods are incremental.
>
> **Q3**: Moreover, I wonder if it would work equally well to simply use a different LLM to generate those similar pairs for augmentation.
>
> **A3**: We use different LLM paraphrases to generate augmented data for ConvexDA, and the results are updated in the Appendix I. We apply three different paraphrasers, including T5-based, Falcon-based, and Parrot-based paraphrases, as shown in Table 13, the choice of paraphraser in ConvexDA does not substantially influence the effectiveness of ConvexDA.
>
> **Q4**: Please compare to methods that simply use LLMs for data augmentation.
>
> **A4**: In Appendix F, we present a comparison between ConvexDA and standard data augmentation (DA) techniques, such as paraphrasing. ConvexDA is not aimed at outperforming traditional data augmentation methods in terms of raw performance. Figure 8 shows that ConvexDA achieves the desired augmentation effect with greater efficiency, making it a superior choice when considering the trade-offs inherent in the data augmentation process.
>
> If you have any other questions, please feel free to ask.

---

> > ### Author Response · Authors · 2023-11-21
> >
> > Dear Reviewer,
> >
> > Please feel free to reach out if you have any additional questions or would like any clarification. We'll do our best to respond in the remaining 40 hours before the discussion period ends.

---

### Official Review · Reviewer_5JFN · 2023-11-10

**Soundness:** 1 poor
**Presentation:** 2 fair
**Contribution:** 3 good
**Rating:** 6
**Confidence:** 4

**Summary:**

The authors study reward models that are used for RLHF  to tune LLMs for desirable generations. Specifically the aspect of (in-)consistency is studied. The paper studies the following research questions:

- How to measure in-consistency of reward models? :
  - The authors introduce the Contrast Instructions benchmarking strategy to measure in-consistency.
  - The benchmark consists of quadruplets consisting of lexically similar instructions but different responses. Two metrics are proposed:
    - Response consistency: Can the RM assign a higher score to the correct response given the instruction
    - Instruction consistency: Can the RM assign a higher score to the correct instruction given the response.
  - The benchmarking is automatically constructed using existing open source human preference datasets. A sentence embedding model SimSE is used to find pairs of instructions that are lexically similar, but semantically different.
  - This strategy has been explored with 4 popular open source human preference datasets.
  - They find a huge performance gap on this benchmark between human judgements and reward models trained with the 7B LLaMa checkpoint
- How to reduce the gap and improve consistency of reward models? : Two techniques ConvexDA and RewardFusion are introduced that can be incorporated into the training and inference stage of reward models at no additional computation cost. The authors show this helps improve consistency.
  - ConvexDA: At training time, the data is augmented by substituting words in the responses with synonyms generated using WordNet
  - RewardFusion:  At inference time, a weighted average reward score between similar training examples and given instruction response pair is used
- How does reward inconsistency influence downstream performance of chatbots after RLHF? : The authors show that using a more consistent RM for RLHF can lead to more preferable downstream generations.

**Strengths:**

- The paper proposes a new way of evaluating reward models and focuses on the new important aspect of consistency of reward models. It also highlights how existing reward model evaluation methods do not capture this aspect.
- The benchmark creation for evaluation is intuitive and automatic and can be easily applied to any existing instruction tuning dataset
- The paper highlights the importance of using more consistent reward models for RLHF
- The paper also propose simple methods that show slight improvements in performance on the proposed evaluation metric

**Weaknesses:**

- Contrast instructions benchmark: To sample lexically similar but semantically different pairs of instructions the authors only sample instruction pairs that lie within the similar range of 0.75 and 0.9. It would be useful to explain the sensitivity of this hyperparameter and study how well this prevents sampling semantically similar instructions.
- When evaluating existing RM on this benchmark the reward models get a low C_res score of 53.6, even though C_res conceptually resembles the RM learning objective. This is surprising given the benchmark was constructed using the same datasets used for training. It would be useful to verify if appropriate hyper-parameter tuning was performed and if the model is able to overfit and get a high C_res score.
- Limited models and scales: It would be useful to explore more pretrained models and perform a model and dataset scaling analysis to measure if and how the consistency of reward models varies across different models and scales and if similar observations are found. Would be useful to do a similar analysis when evaluating the impact of the proposed improvement methods ConvexDA and RewardFusion to see how well they perform with different models.
- It would be useful to explain how the proposed data augmentation and inference time technique are designed to help with improving the performance in contrast instructions benchmark. This is especially important given the very small improvement on the Contrast Instructions benchmark with the proposed approaches.
- The two methods ConvexDA and RewardFusion that are proposed are not used to build more consistent reward models that are used during the RLHF stage. Instead fine tuning on contrast instructions format is used. However, in appendix C, it is mentioned the “more consistent” reward model is equipped with ConvexDA.
  - It would be useful to evaluate models that have been trained (during RLHF) using reward models that are equipped with the proposed strategies (ConvexDA and RewardFusion) to show the improvements brought by this approach
  - Please clarify what “finetuning with contrast instructions format” consists of
  - Please clarify if ConvexDA was used to train the reward models for this stage as mentioned in the appendix.

**Questions:**

- Could you please explain how the proposed approaches are designed to help address the inconsistency in the reward model?
- Could you explain the reason behind the very low C_res score of the trained reward model on the same dataset, even though C_res resembles the RM training objective?
- Could you explain what finetuning on contrast instruction format dataset involves?

---

> ### Author Response · Authors · 2023-11-19
>
> Thank you for your insightful comments and questions! We’ve made a revision to the draft and updated results that could address some of your comments.
>
> **Q1**: The authors only sample instruction pairs that lie within the similar range of 0.75 and 0.9. It would be useful to explain the sensitivity of this hyperparameter and study how well this prevents sampling semantically similar instructions.
>
> **A1**: We hand-crafted our cosine similarity range for the following reasons:
> - Upper Bound: We set the upper limit (e.g., 0.9) to avoid retrieving identical instructions. This is crucial to ensure  that $I_{A} \circ r_{A}$ is better than $I_{B} \circ r_{A}$.
>
> - Lower Bound: We avoided setting the lower bound too low (e.g., 0.5) to maintain the challenge for the RM. A lower range would make it too easy for the RM to judge, failing to test whether RM has a nuanced understanding of human preferences.
>
> - Range Interval: We didn’t opt for a narrow range (e.g., [0.85, 0.9]) as it would greatly limit the size of our Contrast Set.
> To satisfy all of three factors, we can only handcraft a proper range.
>
> In Appendix G, we include a human analysis on each of the dataset, to see how well humans can do in terms of $C_{res}$ and $C_{ins}$. On average over 4 datasets , we see that humans rank responses correctly ~82% of the time. If we allow for Google search to access relevant knowledge, the accuracy goes up to 96%. We think the gap between RM vs. human performance helps justify our observations on RM inconsistency, as well as contrast instructions as a benchmarking strategy.
>
> **Q2**: When evaluating existing RM on this benchmark the reward models get a low C_res score of 53.6, even though C_res conceptually resembles the RM learning objective. This is surprising given the benchmark was constructed using the same datasets used for training. It would be useful to verify if appropriate hyper-parameter tuning was performed and if the model is able to overfit and get a high C_res score.
>
> **A2**: We want to clarify that – $C_{res}$ only **conceptually** resembles the RM training objective, as in, both involve distinguishing between two responses given an instruction. In practice, however, RM training data is usually constructed using “proxies” of human preference labels. For example, within the open-source community (e.g. w/ Llama), examples from StackExchange make up for the majority of the RM training data.  StackExchange follows a specific “proxy” way of defining human preferences, e.g. within the same question, each human preferences example is constructed from responses to the same question/thread, and responses with the highest votes are considered the preferred responses. What we intend to show with our $C_{res}$ metric is that RMs trained this way only fits the training distribution, and don’t generalize to what we think RM should be doing – i.e. being able to distinguish between good/bad responses consistently. With contrast instructions, we observe that RMs fail to do so even when the questions come from the domain (e.g. StackExchange) as seen in training.
>
> To your concern about our implementation – In our experiments, we actually tested on an **off-the-shelf** open-source RM trained on StackExchange data– Stack-LLaMa from the Hugging Face RLHF team (https://huggingface.co/trl-lib/llama-7b-se-rm-peft). The results are shown in Table 3. We see close to random performance from the off-the-shelf RM under $C_{res}$ as well.
>
> **Q3**: Limited models and scales
>
> **A3**: Thanks for your suggestion – In Appendix H, we've added experiments with models of different sizes. We observe that scaling model size leads to (1) increased test performance on the original human preference dataset, but (2) the $C_{res}$ and $C_{ins}$ performance remains flat. This echoes our hypothesis that – standard RM training fits only the training distribution, but does not lead to a generalized model that can distinguish good/bad responses consistently.
>
> **Q4**: The two methods ConvexDA and RewardFusion that are proposed are not used to build more consistent reward models that are used during the RLHF stage. Instead fine tuning on contrast instructions format is used. However, in appendix C, it is mentioned the “more consistent” reward model is equipped with ConvexDA.
>
> **A4**: We apologize for a typo in Appendix C related to our experimental setup. In our research, we fine-tuned the consistent RM with our contrast set without ConvexDA. Using ConvexDA and RewardFusion to improve the RM for RLHF didn't significantly improve the chatbot's performance due to their limited improvements in enhancing RM consistency. Since we focus on the trickle-down effect of RM consistency, we select the most effective method (training on Contrast Instruction) in RLHF to better observe such effects.

---

> ### Author Response · Authors · 2023-11-19
>
> **Q5**: Please clarify what “finetuning with contrast instructions format” consists of.
>
> **A5**: The Contrast Instruction format is identical to the original benchmark, where each pair has an "Instruction" and "response" joined together. We've corrected this writing to clear up any potential confusion.
>
> If you have any other questions, please feel free to ask.

---

> > ### Author Response · Authors · 2023-11-21
> >
> > Dear Reviewer,
> >
> > Please feel free to reach out if you have any additional questions or would like any clarification. We'll do our best to respond in the remaining 40 hours before the discussion period ends.

---

> > > ### Comment · Reviewer_5JFN · 2023-12-02
> > >
> > > Thank you for the clarifications and updating the paper. All the responses have addressed my questions. I looked at the new Appendix G and H. It is great to see the new results across model scales. While it is interesting to see that the C_res and C_ins performance remains flat, I would like to note it would be useful to compare different models belonging to the same suite of models, such that they are trained similarly on similar data. Would highly recommend doing this analysis using the Pythia suite of models if possible.
> > >
> > > Overall, this work brings to attention an important aspect of reward modelling. I have raised my review score.

---

### Author Response · Authors · 2023-11-19
**A general response to reviewers**

Dear reviewers, we sincerely appreciate your efforts, here is a general response based on our updated version:

**The inconsistency exposed by our Contrast Instruction is a cross-task, cross-model issue.**

- Cross-task: Our experiments involved multiple tasks, and it was observed that standard-trained reward models (RM) consistently exhibit severe inconsistency issues across each task.

- Cross-model: Appendix H details experiments conducted on various models, with the findings summarized in Table 12. These results reinforce the presence of inconsistency issues in a model-agnostic manner

**Implementation questions**:

We provide two compelling evidence points to address potential concerns about our implementation influencing these results.
- First, we included an open-source RM from the Huggingface RLHF team, used on Stack-LLaMa, which also demonstrated severe performance issues when assessed with our Contrast Instruction, including random guessing performance. This shows that our experimental setup does not cause the issue.
- Second, we have rigorously optimized our experimental configurations across several iterations to ensure they are as close to optimal as possible. This is evidenced by our RMs achieving an average accuracy of 80% on their original tests, as shown in Table 8, suggesting that the inconsistency issue is not due to suboptimal experimental settings but is a fundamental issue with the standard RM modeling.

**Major updates**: the updated contents are highlighted in purple color, including the following aspects.
- Appendix G: We use both human and automatic evaluation (GPT-4) to double-check the quality of our Contrast Instruction.
- Appendix H: We show RM inconsistency is a general issue across different models.
- Appendix I: We use different paraphrasers for our ConvexDA, and the performance remains stable.
- Appendix J: We invite more human annotators to evaluate the trickle-down effect of RM consistency, and show the Copen correlation between two groups of human evaluation.

**An access to our data**

In fact, our Contrast Instructions are more difficult than the cases shown in Table 1. The selections in Table 1 and Figure 1 were specifically chosen for clarity and to facilitate an easier understanding of what constitutes a Contrast Instruction. Now, we have uploaded the train split of our Contrast Instructions in the system.

---

### Meta-Review · Area_Chair_pw74 · 2023-12-05

**Metareview:**

The paper investigates reward model (RM) inconsistency in RLHF, introducing "Contrast Instructions" as a new benchmarking strategy
that pairs lexically similar instructions (identified using an embedding model) with different ground truth responses. It reveals significant inconsistencies relative to humans in standard-trained RMs across tasks and models and show how this inconsistency negatively impact RLHF model training.  The paper's strengths include its its focus on an underexplored aspect of RM training and easy to use dataset creation strategy for benchmarking / training. Additionally, the paper is well-written and easy to read.


The weaknesses which remain unaddressed by the authors mainly include the limited scaling experiments, particularly in terms of model sizes beyond 7B. Additionally, the proposed methods ConvexDA and RewardFusion seem to be of limited significance and could be moved to the appendix.

**Justification For Why Not Higher Score:**

Most of reviewers raised their scores after discussion with the authors but the paper is still borderline with most scores marginally above or below acceptance. Overall, I tend to agree with instead of adding more "novel" contributions, the authors could keep their paper focused on "Contrast Instructions". Nevertheless, I urge the reviewers to further improve their camera ready version to incorporate reviewer feedback.

**Justification For Why Not Lower Score:**

\I read the paper and the discussion and believe that authors have done a good job in addressing most of the reviewers' concerns.

---

### Decision · Program_Chairs · 2024-01-16

Accept (poster)